# SimPER: A Minimalist Approach to Preference Alignment without Hyperparameters

**Teng Xiao**[1*], **Yige Yuan**[2*], **Zhengyu Chen**[3], **Mingxiao Li**[4],
**Shangsong Liang**[5], **Zhaochun Ren**[6], **Vasant G Honavar**[1]
[1]Pennsylvania State University [2]University of Chinese Academy of Sciences
[3]Meituan Inc [4]Tencent AI Lab [5]Sun Yat-Sen University [6]Leiden University
tengxiao@psu.edu, yuanyige923@gmail.com, vhonavar@psu.edu

## Abstract

Existing preference optimization objectives for language model alignment require additional hyperparameters that must be extensively tuned to achieve optimal performance, increasing both the complexity and time required for fine-tuning large language models. In this paper, we propose a simple yet effective hyperparameter-free preference optimization algorithm for alignment. We observe that promising performance can be achieved simply by optimizing inverse perplexity, which is calculated as the inverse of the exponentiated average log-likelihood of the chosen and rejected responses in the preference dataset. The resulting simple learning objective, `SimPER` (**Sim**ple alignment with **Per**plexity optimization), is easy to implement and eliminates the need for expensive hyperparameter tuning and a reference model, making it both computationally and memory efficient. Extensive experiments on widely used real-world benchmarks, including MT-Bench, AlpacaEval 2, and **10** key benchmarks of the Open LLM Leaderboard with **5** base models, demonstrate that `SimPER` consistently and significantly outperforms existing approaches—even without any hyperparameters or a reference model. For example, despite its simplicity, `SimPER` outperforms state-of-the-art methods by up to **5.7** points on AlpacaEval 2 and achieves the highest average ranking across **10** benchmarks on the Open LLM Leaderboard. The source code for `SimPER` is publicly available at the Github: https://github.com/tengxiao1/SimPER.

## 1 Introduction

Learning from preference data plays a crucial role in fine-tuning large language models to ensure that pretrained LLMs are aligned with human or societal values and preferences (Bai et al., 2022; Ouyang et al., 2022; Stiennon et al., 2020). In recent years, reinforcement learning from human feedback (RLHF) (Ouyang et al., 2022; Christiano et al., 2017) has been proposed for fine-tuning language models based on human preferences. In the RLHF pipeline (Ouyang et al., 2022), a reward model is first fit to a dataset of human preferences in the form of a classifier between chosen and rejected responses. Next, an LLM policy is trained using RL algorithms such as proximal policy optimization (PPO) (Schulman et al., 2017) to generate responses given the input prompts with high reward.

While RLHF produces models with impressive capabilities across diverse tasks, ranging from programming to creative writing, it introduces notable complexities into the training process (Engstrom et al., 2020; Rafailov et al., 2024), involving inefficient and unstable optimization, as well as training on separate reward and policy models. This potentially worsens the sample complexity and compromises efficient convergence. To address these issues, offline preference fine-tuning (Tajwar et al., 2024) methods, such as DPO (Rafailov et al., 2024), IPO (Azar et al., 2024), and KTO (Ethayarajh et al., 2024a), have been proposed to replace RLHF with supervised learning on human preference data. More recently, SimPO (Meng et al., 2024) eliminates the need for a reference model, making DPO more compute and memory efficient. These methods eliminate the need for explicit reward modeling by directly using the *likelihood* of language model policy to define a *implicit reward* fitted to the preference data, while achieving notable competitive performance (Tajwar et al., 2024).

---

*Equal contribution.

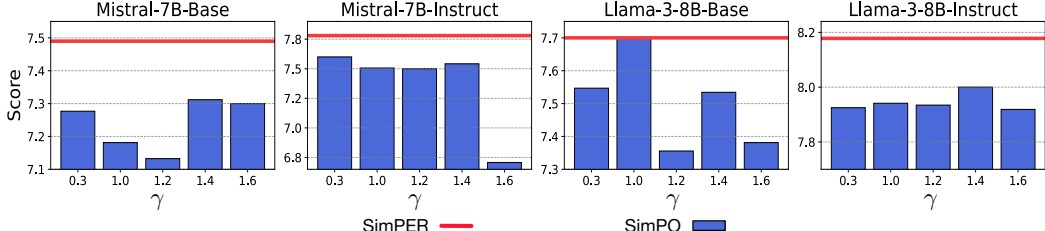

Figure 1: Evaluation on the MT-Bench Score (1-10) of `SimPO` and our `SimPER` across different large language models reveals the high sensitivity and instability of `SimPO` with respect to its hyperparameter $\gamma$ across models. In contrast, our `SimPER`, which operates without any hyperparameters in the objective function, consistently and significantly outperforms `SimPO` across a wide range of models. Additional experimental evidence on other widely used benchmarks is provided in Section 4.

However, these methods require additional hyperparameters that must be carefully tuned as shown in Table 1, and the performance of current preference optimization methods, such as DPO, KTO, IPO, and others, is highly sensitive to these hyperparameters across different LLMs, as already shown by (Hugging-Face, 2023; Liu et al., 2024a; Meng et al., 2024; Liu et al., 2024c; Wu et al., 2024). In Figure 1, we also show that tuning hyperparameters is also crucial to achieve optimal performance with the recent state-of-the-art

Table 1: State-of-the-art preference fine-tuning losses are compared in terms of properties: hyperparameters, the number of hyperparameters, and the elimination of the reference model (see Appendix B.2 for details of these methods and hyperparameters).

| Method | Hyperparameters | #Hyperparameters | w/o Reference Model |
|---|---|---|---|
| DPO | $\beta$ | 1 | ✗ |
| IPO | $\beta$ | 1 | ✗ |
| KTO | $\lambda_l, \lambda_w, \beta$ | 3 | ✗ |
| CPO | $\lambda, \beta$ | 2 | ✓ |
| SLiC | $\delta, \lambda$ | 2 | ✓ |
| SimPO | $\gamma, \beta$ | 2 | ✓ |
| SimPER | - | 0 | ✓ |

algorithm `SimPO`, which eliminates the need for a reference model. This challenge largely prevents us from aligning large language models in real-world applications, given that a single post-training process for alignment is usually very expensive and takes a long time (Dubey et al., 2024). To this end, we ask an important research question for large language model alignment: *Can we design an efficient and effective hyperparameter-free preference optimization method for alignment?*

In this paper, we answer this question affirmatively. We propose `SimPER` (**Sim**ple alignment with **Per**plexity optimization), a simple yet effective offline preference optimization objective that eliminates the need for a reference model and any tunable hyperparameters. The key to `SimPER` is directly optimizing the reverse perplexity of chosen and rejected responses within the preference dataset. Perplexity (Jelinek et al., 1977) is a well-known evaluation metric for language modeling, commonly used to assess a model's ability to process long text. It is calculated as the inverse of the exponentiated average log-likelihood of the responses. `SimPER` achieves alignment by solving a reverse perplexity optimization problem, minimizing perplexity over the chosen response while maximizing perplexity over the rejected response, enabling the model to better align with human preferences. Our simple `SimPER` validates that perplexity is also an effective optimization indicator for LLM alignment.

Moreover, our further analysis proves that, unlike optimizing Kullback-Leibler divergence (KLD) in `SimPO`, our algorithm effectively minimizes the Total Variation distance (TVD). From a gradient perspective, the robust nature of TVD balances gradients from positive and negative responses, which ensures that the contribution from negative samples does not overshadow those from its positive counterpart, thereby mitigating the issue of decreasing the likelihood of chosen response during preference optimization as noticed by recent works (Meng et al., 2024; Pal et al., 2024).

We empirically demonstrate that `SimPER` enjoys promising performance on extensive benchmarks such as the Open LLM Leaderboard (Beeching et al., 2023a), MT-Bench (Zheng et al., 2023), and AlpacaEval 2 (Li et al., 2023b). Despite its simplicity, our results show that `SimPER` consistently and significantly outperforms existing approaches across various large language models, without the need for any hyperparameters and a reference model in the objective function for alignment.

## 2 RELATED WORK

Reinforcement Learning from Human Feedback (RLHF) is an effective technique designed to align LLMs with human preferences (Christiano et al., 2017). The training process for RLHF includes

three stages: initial supervised fine-tuning (Zhou et al., 2024; Xia et al., 2024), training of the reward model from human preference data (Gao et al., 2023a; Xiao & Wang, 2021), and policy optimization using reinforcement learning, notably Proximal Policy Optimization (PPO) (Schulman et al., 2017). While RLHF provides significant benefits in various domains, such as instruction-following (Ouyang et al., 2022), safety alignment (Bai et al., 2022), and truthfulness enhancement (Tian et al., 2023), it requires a more complex training pipeline compared to traditional supervised learning methods.

Recent literature highlights the inherent complexity of online preference optimization algorithms, driving the exploration of more efficient offline alternatives. A notable advancement is Direct Preference Optimization (DPO) (Rafailov et al., 2024), which eliminate the need for explicit reward modeling by directly using the likelihood of policy to define an implicit reward fitted to the preference data. Inspired by DPO, various methods such as IPO (Azar et al., 2024), KTO (Ethayarajh et al., 2024b), and others (Yuan et al., 2024a; Xu et al., 2024; Hong et al., 2024; Xiao et al., 2024b;a) have been proposed. While these approaches are effective, they typically necessitate an extensive search for one or more hyperparameters, as well as the use of a reference model. Recently, SimPO (Meng et al., 2024) removed the need for a reference model in DPO, yet introduced two additional hyperparameters: the reward scaling factor and the target reward margin, which require significant manual tuning. Tuning hyperparameters often entails an iterative trial-and-error process, resulting in substantial computational overhead, particularly for large-scale language models. In this paper, we introduce `SimPER`, a simple yet effective objective that eliminates the need for costly hyperparameter tuning and a reference model, thus enhancing both learning and memory efficiency in practice.

Also worth mentioning is a body of work on perplexity (Jelinek et al., 1977) in language modeling. Researchers use perplexity, an evaluation metric aligned with the causal language modeling objective of LLMs, to assess whether a test input falls within the LLM's expertise and how it relates to the LLM's pretraining data (Chen et al., 2024b; Marion et al., 2023; Gonen et al., 2023). A lower perplexity indicates that the model's predictions are generally more accurate, while a higher perplexity suggests that the model finds the content more unpredictable. Recent work (Ankner et al., 2024; Muennighoff et al., 2024) also uses perplexity to identify high-quality subsets of large-scale text datasets that improve performance. Perplexity has also been used to fuse knowledge from multiple models (Mavromatis et al., 2024). As perplexity provides a sequence-length normalized expression of the model's confidence, recent works have utilized the inverse perplexity score to detect hallucinations (Valentin et al., 2024) and for confidence estimation (Liu et al., 2024b). In contrast to these works, we propose a simple yet effective alignment objective based on perplexity, demonstrating that perplexity is also a surprisingly effective indicator for achieving alignment on preference data.

## 3 THE PROPOSED METHOD

### 3.1 BACKGROUND

**Notations**. We consider the problem of preference fine-tuning: Let the text sequences $\mathbf{x} = [x_1, x_2, \ldots]$ denote the prompt, and $\mathbf{y}_w = [y_1, y_2, \ldots]$ and $\mathbf{y}_l = [y_1, y_2, \ldots]$ denote two responses, sampled from the reference policy $\pi_{\text{ref}}(\mathbf{y} \mid \mathbf{x})$. The response pairs are then presented to an oracle who express preferences for responses given the prompt, denoted as $\mathbf{y}_w \succ \mathbf{y}_l \mid \mathbf{x}$, where $\mathbf{y}_w$ and $\mathbf{y}_l$ denote chosen and rejected responses, respectively. Given dataset $\mathcal{D}$, containing preference $(\mathbf{x}, \mathbf{y}_w, \mathbf{y}_l)$, the goal is to learn a language model policy $\pi_{\boldsymbol{\theta}}(\mathbf{y} \mid \mathbf{x})$ parameterized by $\boldsymbol{\theta}$ for aligning human preference.

**DPO**. DPO (Rafailov et al., 2024) is one of the most popular offline preference optimization methods. Instead of learning an explicit reward model like RLHF, DPO uses the log-likelihood of the policy to implicitly represent the reward function via a closed-form expression with the optimal policy:

$$r(\mathbf{x}, \mathbf{y}) = \beta \left( \log \pi_{\boldsymbol{\theta}}(\mathbf{y} \mid \mathbf{x}) - \log \pi_{\text{ref}}(\mathbf{y} \mid \mathbf{x}) \right) + \beta \log Z(\mathbf{x}). \tag{1}$$

DPO aims to optimize $\pi_{\boldsymbol{\theta}}$ based on the Bradley-Terry (BT) preference model (Bradley & Terry, 1952), $p\left(\mathbf{y}_w \succ \mathbf{y}_l \mid x\right) = \sigma\left(r(\mathbf{x}, \mathbf{y}_w) - r(\mathbf{x}, \mathbf{y}_l)\right)$, and with the following maximum likelihood objective:

$$\mathcal{L}_{\text{DPO}}(\boldsymbol{\theta}; \mathcal{D}) = \mathbb{E}_{(\mathbf{x}, \mathbf{y}_w, \mathbf{y}_l) \sim \mathcal{D}} \left[ -\log \sigma(\beta \log \frac{\pi_{\boldsymbol{\theta}}(\mathbf{y}_w \mid \mathbf{x})}{\pi_{\text{ref}}(\mathbf{y}_w \mid \mathbf{x})} - \beta \log \frac{\pi_{\boldsymbol{\theta}}(\mathbf{y}_l \mid \mathbf{x})}{\pi_{\text{ref}}(\mathbf{y}_l \mid \mathbf{x})}) \right], \tag{2}$$

where $\beta$ is a tunable hyperparameter controlling the deviation from the reference model.

**IPO**. The Identity Preference Optimization (IPO) (Azar et al., 2024) also avoids a reward learning process and potentially unstable RL training. Specifically, IPO chooses to directly minimize the

following squared loss regression problems by defining an alternative reward function:

$$\mathcal{L}_{\text{IPO}}(\boldsymbol{\theta}; \mathcal{D}) = \mathbb{E}_{(\mathbf{x}, \mathbf{y}_w, \mathbf{y}_l) \sim \mathcal{D}} \left[ \left( \log \frac{\pi_{\boldsymbol{\theta}}(\mathbf{y}_w \mid \mathbf{x}) \pi_{\text{ref}}(\mathbf{y}_l \mid \mathbf{x})}{\pi_{\boldsymbol{\theta}}(\mathbf{y}_l \mid \mathbf{x}) \pi_{\text{ref}}(\mathbf{y}_w \mid \mathbf{x})} - \frac{1}{2\beta} \right)^2 \right], \tag{3}$$

where $\beta$ is also a hyperparameter. A potential advantage of IPO over DPO is that these methods don't assume a specific preference model, like BT, and can work with general preference probabilities.

**SimPO**. Simple Preference Optimization (SimPO) (Meng et al., 2024) has recently been proposed to eliminate the need for a reference model in DPO while achieving promising performance. SimPO optimizes the length-regularized probability of response pairs with a margin based on the BT model:

$$\mathcal{L}_{\text{SimPO}}(\boldsymbol{\theta}; \mathcal{D}) = \mathbb{E}_{(\mathbf{x}, \mathbf{y}_w, \mathbf{y}_l) \sim \mathcal{D}} \left[ -\log \sigma \left( \frac{\beta}{|\mathbf{y}_w|} \log \pi_{\boldsymbol{\theta}}(\mathbf{y}_w \mid \mathbf{x}) - \frac{\beta}{|\mathbf{y}_l|} \log \pi_{\boldsymbol{\theta}}(\mathbf{y}_l \mid \mathbf{x}) - \gamma \right) \right], \tag{4}$$

where $\gamma$ is an additional hyperparameter indicating a target reward margin. In practice, minimizing the above preference fine-tuning objective, or any other contrastive objectives such as those of KTO (Ethayarajh et al., 2024b) and SLiC (Zhao et al., 2023) (see Appendix B.2), requires extensive hyperparameter tuning. These hyperparameters (e.g., $\beta$ and $\gamma$) play a critical role in alignment performance, as shown by (Bai et al., 2022; Liu et al., 2024a; Meng et al., 2024) (also see Figure 1), and need to be manually adjusted to achieve optimal performance, significantly increasing complexity and time cost. In this paper, we address this limitation by proposing a simple yet effective alignment objective, SimPER, which eliminates the need for a reference model and any tunable hyperparameters required by previous work, making the alignment process on large language model more efficient.

## 3.2 THE LEARNING OBJECTIVE OF SIMPER

In this section, we elaborate on SimPER. The key idea behind SimPER is to encourage the model to minimize the perplexity of the chosen response while simultaneously maximizing the perplexity of the rejected response within the preference dataset. Specifically, we optimize the inverse perplexity, which is calculated as the inverse of the exponentiated average negative log-likelihood of the response. The average log-likelihood of the response under the policy model $\pi_{\theta}$ is defined as follows:

$$r_{\boldsymbol{\theta}}(\mathbf{x}, \mathbf{y}) = \log p_{\boldsymbol{\theta}}(\mathbf{y} \mid \mathbf{x}) = \frac{1}{|\mathbf{y}|} \log \pi_{\boldsymbol{\theta}}(\mathbf{y} \mid \mathbf{x}) = \frac{1}{|\mathbf{y}|} \sum_{i=1}^{|\mathbf{y}|} \log \pi_{\boldsymbol{\theta}} \left( \mathbf{y}_i \mid \mathbf{x}, \mathbf{y}_{<i} \right), \tag{5}$$

where $p_{\boldsymbol{\theta}}(\mathbf{y} \mid \mathbf{x}) = \pi_{\boldsymbol{\theta}}(\mathbf{y} \mid \mathbf{x})^{\frac{1}{|\mathbf{y}|}}$ is the defined geometric mean over the sequence of token probabilities. The perplexity is defined as the exponentiated its average negative log-likelihood as:

$$\text{Perplexity}(\mathbf{y} \mid \mathbf{x}) = \exp\left( -r_{\boldsymbol{\theta}} \left( \mathbf{y} \mid \mathbf{x} \right) \right) = \exp\left( -\frac{1}{|\mathbf{y}|} \log \pi_{\boldsymbol{\theta}}(\mathbf{y} \mid \mathbf{x}) \right), \tag{6}$$

and serves as a metric closely tied to the causal language modeling objective, allowing us to assess whether a new input aligns with the model's knowledge and how it corresponds to the pretraining data (Jelinek et al., 1977; Marion et al., 2023; Gonen et al., 2023). Leveraging perplexity as a measurement of how well a language model predicts the response given the prompt, we formulate the alignment with preference data as an optimization problem that does not require any hyperparameters and a reference model during training. Formally, our SimPER learning objective is given as follows:

$$\mathcal{L}_{\text{SimPER}}(\boldsymbol{\theta}; \mathcal{D}) = -\text{Perplexity}^{-1}(\mathbf{y}_w \mid \mathbf{x}) + \text{Perplexity}^{-1}(\mathbf{y}_l \mid \mathbf{x}) \tag{7}$$

$$= -\exp\left( \frac{1}{|\mathbf{y}_w|} \log \pi_{\boldsymbol{\theta}}(\mathbf{y}_w \mid \mathbf{x}) \right) + \exp\left( \frac{1}{|\mathbf{y}_l|} \log \pi_{\boldsymbol{\theta}}(\mathbf{y}_l \mid \mathbf{x}) \right), \tag{8}$$

where we directly optimize the reverse perplexity of the chosen and rejected response as the reverse perplexity, i.e., the geometric mean over the sequence of token probabilities, effectively quantifies the model confidence as shown in (Valentin et al., 2024; Liu et al., 2024b). Intuitively, SimPER increases the likelihood of the chosen response and decreases the likelihood of rejected response by optimizing the reverse perplexity, effectively aligning the language model with the preference data.

In summary, SimPER employs a simple yet effective formulation that directly aligns with the perplexity generation metric, eliminating the need for a reference model. We empirically find that SimPER still achieves a strong performance without requiring any hyperparameters, unlike previous methods.

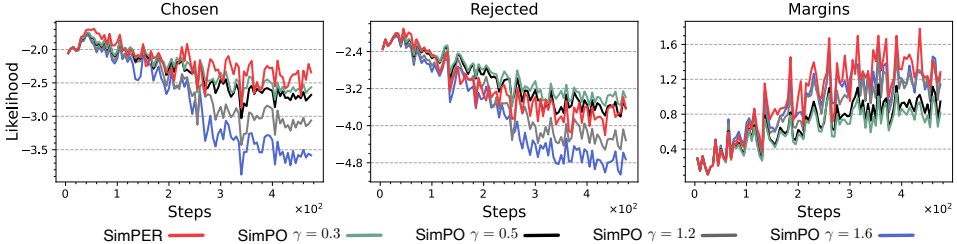

Figure 3: The training dynamics during training of `SimPER` and `SimPO` with different hyperparameters on the Mistral-7B (Results on Llama3-8B can be found in Section 4.2). We can observe that `SimPER` exhibits the least decline in chosen likelihoods, while still achieving the most significant increase in likelihood margins of rejected and chosen, compared to `SimPO` across various hyperparameters.

### 3.3 Analysis and Discussion

In this section, we provide a gradient and divergence analysis to further understand our `SimPER`,

**Gradient Analysis.** We examine the gradients of `SimPER` and the state-of-the-art method `DPO` (Rafailov et al., 2024) and `SimPO` (Meng et al., 2024) and to glean some insight into the optimization process. Note that our analysis also holds for other methods such as IPO (Azar et al., 2024) and SLiC (Zhao et al., 2023). One advantage of the `SimPER` framework is that the gradients of both chosen and rejected responses are more balanced, thus, we can prevent the model from overfitting the rejected responses. We first analyze the following gradients of `DPO` and `SimPO`:

$$\nabla_{\boldsymbol{\theta}}\mathcal{L}_{\text{DPO}}(\boldsymbol{\theta};\mathcal{D}) = -\beta\mathbb{E}_{(\mathbf{x},\mathbf{y}_w,\mathbf{y}_l)\sim\mathcal{D}}\left[w_{\boldsymbol{\theta}}\cdot\left(\frac{\nabla_{\boldsymbol{\theta}}\pi_{\boldsymbol{\theta}}(\mathbf{y}_w\mid\mathbf{x})}{\pi_{\boldsymbol{\theta}}(\mathbf{y}_w\mid\mathbf{x})} - \frac{\nabla_{\boldsymbol{\theta}}\pi_{\boldsymbol{\theta}}(\mathbf{y}_l\mid\mathbf{x})}{\pi_{\boldsymbol{\theta}}(\mathbf{y}_l\mid\mathbf{x})}\right)\right] \tag{9}$$

$$\nabla_{\boldsymbol{\theta}}\mathcal{L}_{\text{SimPO}}(\boldsymbol{\theta};\mathcal{D}) = -\beta\mathbb{E}_{(\mathbf{x},\mathbf{y}_w,\mathbf{y}_l)\sim\mathcal{D}}\left[d_{\boldsymbol{\theta}}\cdot\left(\frac{\nabla_{\boldsymbol{\theta}}p_{\boldsymbol{\theta}}(\mathbf{y}_w\mid\mathbf{x})}{p_{\boldsymbol{\theta}}(\mathbf{y}_w\mid\mathbf{x})} - \frac{\nabla_{\boldsymbol{\theta}}p_{\boldsymbol{\theta}}(\mathbf{y}_l\mid\mathbf{x})}{p_{\boldsymbol{\theta}}(\mathbf{y}_l\mid\mathbf{x})}\right)\right], \tag{10}$$

where the weights $w_{\boldsymbol{\theta}} = \sigma(\beta\log\frac{\pi_{\boldsymbol{\theta}}(\mathbf{y}_l|\mathbf{x})}{\pi_{\text{ref}}(\mathbf{y}_l|\mathbf{x})} - \beta\log\frac{\pi_{\boldsymbol{\theta}}(\mathbf{y}_w|\mathbf{x})}{\pi_{\text{ref}}(\mathbf{y}_w|\mathbf{x})})$ and $d_{\boldsymbol{\theta}} = \sigma(\frac{\beta}{|\mathbf{y}_l|}\log\pi_{\boldsymbol{\theta}}(\mathbf{y}_l|\mathbf{x}) - \frac{\beta}{|\mathbf{y}_w|}\log\pi_{\boldsymbol{\theta}}(\mathbf{y}_w|\mathbf{x}) + \gamma)$ represent the gradient weight in `DPO` and `SimPO`, respectively. $p_{\boldsymbol{\theta}}$ is the geometric mean over the sequence of token probabilities defined in Equation (5). It can be seen that the gradient of the model probability weighted by the reciprocal of the model probability on this rejected response. If the rejection likelihood $\pi_{\boldsymbol{\theta}}(\mathbf{y}_l\mid\mathbf{x}) \to 0$ or $p_{\boldsymbol{\theta}}(\mathbf{y}_l\mid\mathbf{x}) \to 0$, the norm of the gradient on rejected response will be large, which leads to a huge step of parameter update towards decreasing the likelihood of rejected response compared to the increasing of the chosen response.

For instance, in DPO, the gradient ratio between the decrease in the probability of rejected responses and the increase in the probability of chosen responses is as follows: $\frac{\pi_{\boldsymbol{\theta}}(\mathbf{y}_w|\mathbf{x})}{\pi_{\boldsymbol{\theta}}(\mathbf{y}_l|\mathbf{x})} \cdot \frac{\nabla_{\boldsymbol{\theta}}\pi_{\boldsymbol{\theta}}(\mathbf{y}_l|\mathbf{x})}{\nabla_{\boldsymbol{\theta}}\pi_{\boldsymbol{\theta}}(\mathbf{y}_w|\mathbf{x})}$, which becomes infinite when $\pi_{\boldsymbol{\theta}}(\mathbf{y}_l\mid\mathbf{x}) \to 0$. A larger gradient ratio leads to a faster reduction in the probability of a rejected response compared to that of a chosen response, resulting in a more pronounced decrease for rejected responses, which explains why `DPO` and `SimPO` tend to push the model to decrease the likelihood of both chosen and rejected responses during training, as shown in Figure 3. This occurs because rejected and chosen responses often share some tokens, leading to a decline in performance on reasoning-heavy tasks, such as math and coding, as demonstrated in several recent studies (Meng et al., 2024; Pal et al., 2024; Pang et al., 2024; Chen et al., 2024a). For comparison, we calculate the gradient of our `SimPER` with respect to $\theta$ using Equation (8):

$$\nabla_{\boldsymbol{\theta}}\mathcal{L}_{\text{SimPER}}(\boldsymbol{\theta};\mathcal{D}) = -\mathbb{E}_{(\mathbf{x},\mathbf{y}_w,\mathbf{y}_l)\sim\mathcal{D}}\left[\nabla_{\boldsymbol{\theta}}p_{\boldsymbol{\theta}}(\mathbf{y}_w\mid\mathbf{x}) - \nabla_{\boldsymbol{\theta}}p_{\boldsymbol{\theta}}(\mathbf{y}_l\mid\mathbf{x})\right], \tag{11}$$

Where the gradient ratio between rejected and chosen responses is constant, it has a smaller norm than SimPO (Equation 10). This means that the unlearning of rejected responses is more conservative, and `SimPER` reduces the gradient imbalance issue between chosen and rejected responses. As shown in Figure 3, `SimPER` effectively prevents the likelihood of chosen responses from decreasing significantly compared to `SimPO`, while still achieving substantial margins between the likelihood of chosen and rejected responses. Our experiments also show that `SimPER` significantly outperforms `SimPO`.

**Divergence Analysis.** We next present a theoretical analysis of `SimPER`, demonstrating its key properties that are advantageous for fine-tuning LLMs with preferences. Recent works (Tajwar et al.,

2024; Xiao et al., 2024b; Ji et al., 2024) identify mode-seeking behavior as a crucial property for preference alignment, as it reduces the likelihood of rejected responses. In what follows, we show that `SimPER` also theoretically promotes mode-seeking behavior by optimizing the Total Variation distance (TVD) between the model distribution $\pi_\theta$ and the distribution of chosen response. For simplicity, we remove the length averaging from `SimPER` for the analysis.

**Theorem 3.1.** *Minimizing `SFT` with respect to $\theta$ is approximately minimizing the KLD between $\pi_\theta$ and the distribution of the chosen response in the preference dataset, while minimizing our `SimPER` is approximately minimizing the TVD.*

$$\min_{\boldsymbol{\theta}} \mathcal{L}_{\text{SFT}} \Rightarrow \min_{\boldsymbol{\theta}} \text{KL}(\pi_{\text{chosen}}(\mathbf{y} \mid \mathbf{x}) \| \pi_{\boldsymbol{\theta}}(\mathbf{y} \mid \mathbf{x})) = \sum_{\mathbf{y} \in \mathcal{Y}} \pi_{\text{chosen}}(\mathbf{y} \mid \mathbf{x}) \log \frac{\pi_{\text{chosen}}(\mathbf{y} \mid \mathbf{x})}{\pi_{\boldsymbol{\theta}}(\mathbf{y} \mid \mathbf{x})} \quad (12)$$

$$\min_{\boldsymbol{\theta}} \mathcal{L}_{\text{SimPER}} \Rightarrow \min_{\boldsymbol{\theta}} \text{TV}(\pi_{\text{chosen}}(\mathbf{y} \mid \mathbf{x}) \| \pi_{\boldsymbol{\theta}}(\mathbf{y} \mid \mathbf{x})) = \frac{1}{2} \sum_{\mathbf{y} \in \mathcal{Y}} \left| \pi_{\text{chosen}}(\mathbf{y} \mid \mathbf{x}) - \pi_{\boldsymbol{\theta}}(\mathbf{y} \mid \mathbf{x}) \right| \quad (13)$$

The proof is provided in Appendix A. This theorem demonstrates that `SimPER` asymptotically optimizes the TVD (Van Handel, 2014; Ji et al., 2023) between the chosen data distributions and the model distribution. In theory, while `SimPER` and `SFT` aim to discover identical optimal policies, achieving this in practice would require full data coverage and infinite computation. These requirements are not met in practice, and hence, the choice of divergences and the optimization procedure affects performance. TVD measures the average absolute difference between $\pi_{\text{chosen}}$ and $\pi_\theta$ in all possible text sequences, and `SimPER` learns to properly allocate its probability mass to best represent the main portion of the distribution of the chosen response, while ignoring outliers. This promotes mode-seeking behavior, which concentrates the probability mass on certain high-reward regions. In contrast, the KLD in `SFT` encourages assigning equal probability to all responses in the dataset, leading to an overestimation of the long tail of the target distribution, as illustrated in Figure 2.

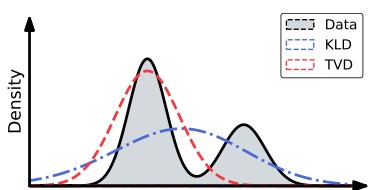

Figure 2: Illustration of the characteristics of KLD and TVD. While `SFT` exhibits mass-covering behavior by minimizing forward KL, `SimPER` exhibits mode-seeking behavior, similar to RLHF (Tajwar et al., 2024), by minimizing TVD.

In summary, forward KLD encourages all chosen responses in datasets to have equal probability, leading to an overestimation of the long tail of the target distribution, whereas reverse TVD sharpens the probability mass on certain high-reward regions of chosen response. Thus, alignment commits to generating a certain subset of high-reward responses, which is more effectively realized by promotes mode-seeking behavior as shown in recent works (Tajwar et al., 2024; Xiao et al., 2024b).

## 4 EXPERIMENTS

**Models.** Following (Meng et al., 2024), we perform alignment with several families of open-source models, Llama3-8B (Base and Instruct.) (Dubey et al., 2024) and Mistral-7B (Base and Instruct.) (Jiang et al., 2023a). We also use Pythia-2.8B (Biderman et al., 2023; Rafailov et al., 2024).

**Datasets.** For the Llama3-8B-Base and Mistral-7B-Base setups, we follow the same training pipeline as Zephyr (Tunstall et al., 2023) and evaluate `SimPER` on the widely used benchmark dataset for preference fine-tuning: the UltraFeedback Binarized dataset (Cui et al., 2023; Tunstall et al., 2023). For the Llama3-8B-Instruct and Mistral-7B-Instruct setups, we evaluate using the on-policy datasets generated by SimPO (Meng et al., 2024). Specifically, we use prompts from UltraFeedback and regenerates the chosen and rejected response pairs with the SFT models. For each prompt, it generates five responses using the SFT model with sampling. We then use llm-blender/PairRM (Jiang et al., 2023b) to score the five responses, selecting the highest-scoring one as the chosen response and the lowest-scoring one as the rejected response. For Pythia-2.8B, Anthropic-HH dataset (Bai et al., 2022) is used for dialogue generation to produce helpful and harmless responses (Rafailov et al., 2024).

**Evaluation benchmarks.** Following previous work (Rafailov et al., 2024; Tunstall et al., 2023), we evaluate methods fine-tuned on the benchmarks on HuggingFace Open LLM Leaderboard v1 and v2 (Gao et al., 2023b) and instruction-following benchmarks (AlpacaEval2, MT-Bench). For evaluation on Anthropic-HH, we use GPT-4 for zero-shot pair-wise evaluation, consistent with human judgments (see prompts in Appendix B.1). The task and evaluation details are given in Appendix B.1.

.

Table 2: AlpacaEval 2 (Li et al., 2023a) and MT-Bench (Zheng et al., 2023) results under the four settings. LC and WR denote length-controlled and raw win rate, respectively. Our `SimPER` can achieve surprisingly good performance without any hyperparameters across various settings.

| Method | Mistral-7B-Base | | | Mistral-7B-Instruct | | | Llama3-8B-Base | | | Llama3-8B-Instruct | | |
|---|---|---|---|---|---|---|---|---|---|---|---|---|
| | AlpacaEval 2 | | MT-Bench | AlpacaEval 2 | | MT-Bench | AlpacaEval 2 | | MT-Bench | AlpacaEval 2 | | MT-Bench |
| | LC (%) | WR (%) | GPT-4 | LC (%) | WR (%) | GPT-4 | LC (%) | WR (%) | GPT-4 | LC (%) | WR (%) | GPT-4 |
| SFT | 8.4 | 6.2 | 6.3 | 17.1 | 14.7 | 7.5 | 6.2 | 4.6 | 6.6 | 26.0 | 25.3 | 8.1 |
| DPO | 15.1 | 12.5 | 7.3 | 26.8 | 24.9 | 7.6 | 18.2 | 15.5 | 7.7 | 40.3 | 37.9 | 8.0 |
| SLiC | 10.9 | 8.9 | 7.4 | 24.1 | 24.6 | **7.8** | 12.3 | 13.7 | 7.6 | 26.9 | 27.5 | 8.1 |
| IPO | 11.8 | 9.4 | 7.2 | 20.3 | 20.3 | **7.8** | 14.4 | 14.2 | 7.4 | 35.6 | 35.6 | **8.3** |
| KTO | 13.1 | 9.1 | 7.0 | 24.5 | 23.6 | 7.7 | 14.2 | 12.4 | **7.8** | 33.1 | 31.8 | 8.2 |
| CPO | 9.8 | 8.9 | 6.8 | 23.8 | 28.8 | 7.5 | 10.8 | 8.1 | 7.4 | 28.9 | 32.2 | 8.0 |
| SimPO | 21.5 | 20.8 | 7.3 | 32.1 | 34.8 | 7.6 | 22.0 | 20.3 | 7.7 | 44.7 | 40.5 | 8.0 |
| SimPER | **22.4** | **21.3** | **7.5** | **37.8** | **39.5** | **7.8** | **25.2** | **22.9** | 7.7 | **48.5** | **45.7** | 8.2 |

Table 3: Evaluation results on various tasks from the Huggingface Open Leaderboard show that our simple yet effective `SimPER` achieves superior or comparable performance to other, more complex preference fine-tuning methods, despite eliminating both hyperparameters and the reference model. Although `SimPO` (Meng et al., 2024) also eliminates the reference model, our `SimPER` demonstrates significant improvements over it across various settings without relying on any hyperparameters.

| | Method | MMLU-PRO | IFEval | BBH | GPQA | MUSR | MATH | GSM8K | ARC | TruthfulQA | Winograd | Avg. Rank |
|---|---|---|---|---|---|---|---|---|---|---|---|---|
| **Mistral-7B Base** | DPO | 26.73 | 10.49 | 43.27 | 28.44 | 43.65 | 1.36 | 21.76 | 61.26 | 53.06 | 76.80 | 4.7 |
| | SLiC | 26.52 | 12.45 | 42.33 | 27.93 | 33.74 | 1.38 | 33.74 | 55.38 | 48.36 | 77.35 | 5.0 |
| | IPO | 25.87 | 11.52 | 40.59 | 28.15 | 42.15 | 1.25 | 27.14 | 60.84 | 45.44 | **77.58** | 5.4 |
| | KTO | 27.51 | 12.03 | 43.66 | 29.45 | 43.17 | 2.34 | **38.51** | 62.37 | **56.60** | 77.27 | 2.5 |
| | CPO | 27.04 | 13.32 | 42.05 | 28.45 | 42.15 | 2.15 | 33.06 | 57.00 | 47.07 | 76.48 | 4.5 |
| | SimPO | 27.13 | 10.63 | 42.94 | 29.03 | 39.68 | 2.49 | 22.21 | 62.63 | 50.68 | 77.54 | 3.8 |
| | SimPER | **27.84** | **15.83** | **43.99** | **30.12** | **43.95** | **2.57** | 33.02 | **63.50** | 53.64 | 76.25 | **2.0** |
| **LLama3-8B Base** | DPO | 31.58 | 33.61 | 47.80 | 32.23 | 40.48 | 4.53 | 38.67 | 64.42 | 53.48 | 76.80 | 4.2 |
| | SLiC | 31.11 | 32.37 | 46.53 | 33.29 | 40.55 | 3.92 | 48.82 | 61.43 | 54.95 | **77.27** | 4.5 |
| | IPO | 30.18 | 31.52 | 46.78 | 32.61 | 39.58 | 4.02 | 22.67 | 62.88 | 54.20 | 72.22 | 6.4 |
| | KTO | 31.16 | 37.10 | 47.98 | 33.72 | 40.21 | 4.14 | 38.97 | 63.14 | 55.76 | 76.09 | 4.0 |
| | CPO | 30.95 | 38.57 | 47.17 | 33.15 | 41.59 | 4.25 | 46.93 | 61.69 | 54.29 | 76.16 | 4.2 |
| | SimPO | 31.61 | 37.55 | 48.38 | 33.22 | 40.08 | 4.23 | 31.54 | 65.19 | 59.46 | 76.32 | 3.4 |
| | SimPER | **31.99** | **41.78** | **48.62** | **33.80** | **46.03** | **4.61** | **51.02** | **67.06** | **62.59** | 76.24 | **1.3** |
| **Mistral-7B Instruct** | DPO | 26.81 | 22.89 | 45.46 | 28.19 | **46.43** | 1.89 | 35.25 | 66.89 | 68.40 | 76.32 | 3.8 |
| | SLiC | 25.69 | 29.53 | 45.24 | 27.04 | 43.90 | 1.95 | 39.65 | 59.90 | 65.30 | 76.32 | 5.3 |
| | IPO | 25.75 | 27.85 | 43.81 | 26.61 | 43.55 | 2.02 | 39.42 | 63.31 | 67.36 | 75.85 | 5.8 |
| | KTO | 27.46 | 35.42 | 45.34 | 28.19 | 45.77 | 2.35 | 38.80 | 65.72 | 68.43 | 75.91 | 3.2 |
| | CPO | 26.85 | 36.81 | 45.01 | 28.15 | 43.28 | 2.28 | 38.74 | 63.23 | 67.38 | **76.80** | 4.4 |
| | SimPO | 27.10 | 37.52 | 45.70 | 28.04 | 44.71 | 2.19 | 35.25 | **66.89** | 68.40 | 76.32 | 3.3 |
| | SimPER | **27.85** | **39.84** | **46.17** | **28.36** | 44.92 | **2.51** | 40.11 | 66.13 | **68.78** | 76.40 | **1.5** |
| **LLama3-8B Instruct** | DPO | 35.86 | 44.57 | 48.31 | 31.04 | 39.02 | 8.23 | 49.81 | **63.99** | 59.01 | 74.66 | 3.0 |
| | SLiC | 33.25 | 44.01 | 47.55 | 30.52 | 38.10 | 8.29 | 66.57 | 61.26 | 53.23 | **76.16** | 4.6 |
| | IPO | 32.97 | 43.27 | 46.31 | 30.95 | 38.58 | 8.02 | 58.23 | 61.95 | 54.64 | 73.09 | 5.5 |
| | KTO | 35.00 | 40.12 | 47.15 | 29.70 | 38.10 | 7.63 | 57.01 | 63.57 | 58.15 | 73.40 | 5.2 |
| | CPO | 34.56 | 44.08 | 48.51 | 30.08 | 38.81 | 7.75 | 67.40 | 62.29 | 54.01 | 73.72 | 4.4 |
| | SimPO | 35.09 | 43.05 | 48.95 | 31.29 | 39.15 | 8.16 | 50.72 | 62.80 | **60.70** | 73.32 | 3.5 |
| | SimPER | **36.68** | **46.06** | **49.51** | **31.71** | **39.35** | **8.99** | **68.61** | 62.37 | 55.71 | 75.72 | **1.7** |

**Baselines.** We compare `SimPER` with the following preference optimization methods: DPO (Rafailov et al., 2024), SLiC (Zhao et al., 2023), IPO (Azar et al., 2024), KTO (Ethayarajh et al., 2024b), CPO (Xu et al., 2024) and SimPO (Meng et al., 2024). As shown in (Meng et al., 2024), hyperparameter tuning is crucial for achieving optimal performance of preference optimization methods. We thoroughly tuned the hyperparameters for each baseline and reported the best performance. More details of baselines and the hyperparameter search space can be found in Appendix B.2.

## 4.1 MAIN RESULTS ON BENCHMARKS

**Results on Instruction-following Benchmarks**. In Table 2, we report the performance on the widely-used instruction-following benchmarks: MT-Bench and AlpacaEval 2. A notable improvement in the performance of `SimPER` on MT-Bench, as well as on AlpacaEval 2, is observed. Notably, the Llama3 fine-tuned by `SimPER` surpassed the performance of best baseline by 4.9 to 5.2 points on the AlpacaEval 2 win rate across base and instruct settings, demonstrating that `SimPER`, while

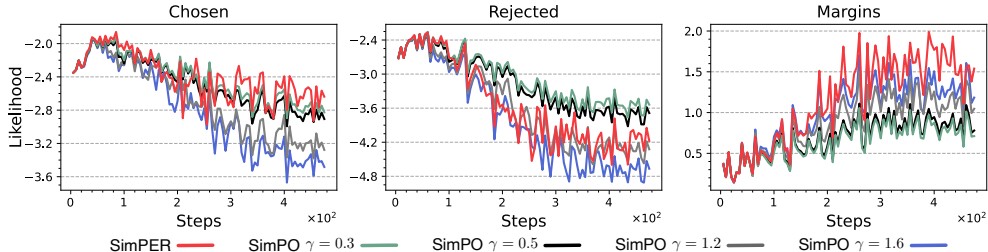

Figure 5: The training dynamics during training of `SimPER` and `SimPO` with different hyperparameters on the Llama3-8B-Base. We can observe that `SimPER` exhibits the least decline in chosen likelihoods, while still achieving the most significant increase in likelihood margins of rejected and chosen, compared to `SimPO` across various hyperparameters, and better performance as shown in Table 2.

eliminating the need for hyperparameters and a reference model, still achieves strong performance. Moreover, `SimPER` consistently achieves superior performance on LC win rate, demonstrating that it can generate high-quality response without substantially increasing response length. We find that MT-Bench exhibits poor separability across different methods, likely due to the limited scale of its evaluation data and its single-instance scoring protocol. This finding aligns with observations reported in (Meng et al., 2024; Li et al., 2024). Nevertheless, on MT-Bench, `SimPER` can still consistently achieves superior or comparable performance across various models. Additionally, we provide examples generated by both `SimPO` and `SimPER` in Appendix C. These examples demonstrate that `SimPER` shows strong promise in aligning language models, ensuring that the generated responses are not only high-quality but also better structured and more reasonable.

**Results on Downstream Tasks**. In Table 3, we present a comparative analysis of performance gains in downstream tasks on the HuggingFace Leadboard, along with a comparative ranking across all tasks. As shown in the table, `SimPER`, despite its simplicity, achieves remarkable improvements over SimPO and DPO, particularly on challenging reasoning benchmarks such as IFEval, MMLU-PRO and Math. Our results demonstrate that `SimPER` is highly effective in improving performance across various models. Notably, `SimPER` outperforms the best baseline by 1.48 to 4.2 points on IFEval across various models. The average improvements over `SimPO` are especially notable in Mistral-Base and Llama3-Base. `SimPER`, despite its simplicity, achieves the best overall performance. These consistent and significant improvements highlight the robustness and effectiveness

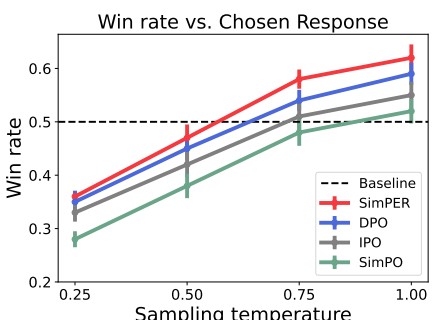

Figure 4: The win rates, computed by GPT-4, in comparison to the chosen responses of test prompts in the Anthropic-HH dataset.

of `SimPER`. In particular, `SimPER` outperforms the recent `SimPO` by 19.48 points in GSM8K and 4.23 points in IFEval on Llama3-Base. On GSM8K, `SimPER` consistently achieves superior performance, though it is occasionally surpassed by CPO on Mistral-Base. We hypothesize that these improvements over SimPO can be attributed to the reduced decrease in the likelihood of chosen responses; As the likelihood of a chosen response decreases, it results in suboptimal performance, particularly in mathematical reasoning and coding tasks, where the chosen responses are very likely ground-truth answers, as also shown in (Pal et al., 2024; Yuan et al., 2024b). These observations suggest the strong potential of `SimPER` in real-world applications due to the elimination of hyperparameters and the reference model in the loss function, making it more computationally and memory efficient.

**Results on Safety Alignment**. The benchmarks used above focus mainly on helpfulness. To further evaluate the effectiveness of `SimPER` in safety alignment on Pythia-2.8B following (Rafailov et al., 2024), we use the Anthropic-HH dataset. Figure 4 shows the win rates computed by GPT-4 over the chosen responses in the test set of Anthropic-HH. Remarkably, `SimPER` aligns better with human preferences than SimPO, DPO, and IPO, achieving win rates of approximately 60% against the chosen responses, indicating that `SimPER` shows strong promise in terms of aligning language models and can ensure that the generated responses are not only high-quality but also safe and harmless.

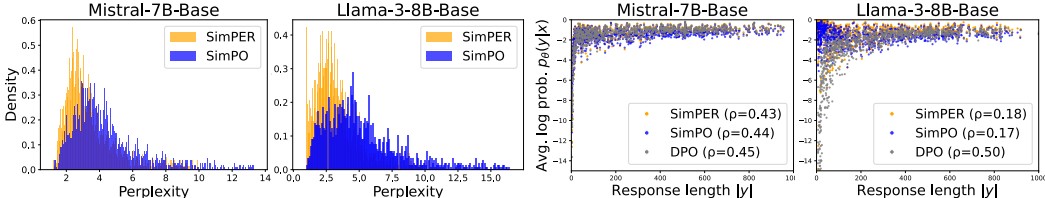

Figure 6: Analyzing perplexity density and response length correlation on Mistral-7B-Base and Llama-3-8B-Base. `SimPER` not only achieves lower perplexity but also does not exploit length bias.

Table 4: Ablation Study Results on Mistral-7B-Base and Llama3-8B-Base: Evaluating the Effects of Removing Length Normalization (`w/o LN`) and Incorporating a Reference Model (`w/ Ref.`).

|  | Method | Open LLM Leaderboard v2 | | | | | | Open LLM Leaderboard v1 | | | |
|---|---|---|---|---|---|---|---|---|---|---|---|
|  |  | MMLU Pro | IFEval | BBH | GPQA | MUSR | MATH | GSM8K | ARC | TruthfulQA | Winograd |
| Mistral-7B Base | SimPER | **27.84** | 15.83 | **43.99** | 30.12 | **43.95** | 2.57 | **31.92** | **63.50** | **53.64** | 76.25 |
|  | - w/o LN | 25.37 | 1.66 | 41.76 | 29.87 | 44.84 | **3.25** | 28.73 | 59.81 | 40.57 | **77.03** |
|  | - w/ Ref. | 27.31 | **16.32** | **43.99** | **30.54** | 40.74 | 2.11 | 26.91 | 61.86 | 43.62 | 76.72 |
| Llama3-8B Base | SimPER | 31.99 | **41.78** | 48.62 | **33.80** | **46.03** | 4.61 | 51.02 | 67.06 | **62.59** | 76.24 |
|  | - w/o LN | 30.08 | 36.31 | 48.62 | 32.05 | 45.37 | 3.17 | 44.28 | 60.58 | 46.87 | **76.64** |
|  | - w/ Ref. | **32.06** | 36.71 | **49.30** | 31.71 | 41.53 | **4.61** | 45.19 | 63.40 | 50.84 | 75.77 |

## 4.2 Ablations and Further Analysis

**Results on Ablation Study**. In Table 4, we evaluated the effects of removing length normalization (`w/o LN`) and incorporating a reference model (`w/ Ref.`) in `SimPER` on the Open LLM Leaderboard. Removing length normalization generally leads to a decline in performance across most tasks, except for a slight improvement in the Winograd. Incorporating a reference model typically results in a performance decrease. However, it does lead to slight improvements in specific tasks such as IFEval, GPQA, and BBH for the Mistral-7B-Base model, and in MMLU Pro, BBH, and MATH for the Llama3-8B-Base model. Based on this ablation study, we can conclude that maintaining length normalization and removing the reference model generally enhances the performance of our method.

**Analysis on Perplexity Density**. As shown in Figure 6, the analysis of perplexity density on the held-out test set indicates that `SimPER` consistently achieves a lower perplexity compared to `SimPO`. Specifically, on Mistral-7B-Base, the density peak of `SimPER` is reduced by approximately 1 relative to `SimPO`, and on the Llama-3-8B-Base, it is reduced by approximately 2. These suggest that `SimPER` is capable of generating more predictable and coherent responses by directly optimizing the perplexity.

**Analysis on Response Length Correlation**. Spearman correlation between response length and log probabilities in Figure 6 shows that `SimPER` exhibits a significantly lower correlation compared to DPO, and is comparable to `SimPO`, suggesting that `SimPER` does not exploit length bias and generate longer sequences. In contrast, `SimPER` results in a spearman correlation coefficient similar to the `SimPO` (Meng et al., 2024). Our results demonstrate that `SimPER`, despite its simplicity, consistently and significantly outperforms existing approaches without substantially increasing response length.

## 5 Conclusion

In this paper, we propose a simple yet effective alignment method for large language models, named `SimPER`. `SimPER` eliminates the need for both hyperparameters and a reference model while achieving strong performance. The key idea of `SimPER` is to directly optimize the reverse perplexity of both the chosen and rejected responses in the preference dataset. Specifically, we minimize the perplexity over the chosen response while maximizing the perplexity over the rejected response, ensuring that the model produces responses with high preference scores. We also provide a theoretical understanding of `SimPER`, demonstrating that it can mitigate the problem of gradient imbalance between the chosen and rejected responses during optimization. Furthermore, we show that `SimPER` implicitly exhibits mode-seeking behavior, leading to strong alignment performance. Extensive experiments on widely used benchmarks demonstrate that `SimPER` significantly outperforms state-of-the-art methods.

ACKNOWLEDGMENT

The work of Vasant G Honavar and Teng Xiao was supported in part by grants from the National Science Foundation (2226025, 2225824), the National Center for Advancing Translational Sciences, and the National Institutes of Health (UL1 TR002014).

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

# A  PROOFS OF THEOREM 3.1

To prove Theorem 3.1 in the main paper, we first present the following Lemmas:

**Lemma A.1.** *Given the model conditional distribution $\pi_\theta(\mathbf{y} \mid \mathbf{x}) = \prod_{i=1}^{|\mathbf{y}|} \pi_\theta\left(\mathbf{y}_i \mid \mathbf{x}, \mathbf{y}_{<i}\right)$ and data distribution $\pi_{\mathrm{data}}(\mathbf{y} \mid \mathbf{x}) = \prod_{i=1}^{|\mathbf{y}|} \pi_{\mathrm{data}}\left(\mathbf{y}_i \mid \mathbf{x}, \mathbf{y}_{<i}\right)$, then we have the following relationship between the sequence-level TVD objective and its token-level factorization:*

$$\mathrm{TV}(\pi_{\mathrm{data}}(\mathbf{y} \mid \mathbf{x}), \pi_\theta(\mathbf{y} \mid \mathbf{x})) \le \mathbb{E}_{\mathbf{y} \sim \pi_{\mathrm{data}}(\mathbf{y}|\mathbf{x})}\left[\sum_{i=1}^{|\mathbf{y}|} \mathrm{TV}(\pi_{\mathrm{data}}^{<i}(\mathbf{y}_i), \pi_\theta^{<i}(\mathbf{y}_i))\right], \quad (14)$$

*where $\pi_{\mathrm{data}}^{<i}(\mathbf{y}_i)$ and $\pi_\theta^{<i}(\mathbf{y}_i)$ are shorts for $\pi_{\mathrm{data}}\left(\mathbf{y}_i \mid \mathbf{x}, \mathbf{y}_{<i}\right)$ and $\pi_\theta\left(\mathbf{y}_i \mid \mathbf{x}, \mathbf{y}_{<i}\right)$, respectively.*

*Proof.* We start by re-writing the TVD loss in the following recursive form:

$$\mathrm{TV}(\pi_{\mathrm{data}}(\mathbf{y} \mid \mathbf{x}), \pi_\theta(\mathbf{y} \mid \mathbf{x})) = \frac{1}{2}\sum_{\mathbf{y} \in \mathcal{Y}}\left|\pi_{\mathrm{chosen}}(\mathbf{y} \mid \mathbf{x}) - \pi_\theta(\mathbf{y} \mid \mathbf{x})\right| \quad (15)$$

$$= \frac{1}{2}\sum_{\mathbf{y}_1, \cdots, \mathbf{y}_{|\mathbf{y}|}}\left|\prod_{i=1}^{|\mathbf{y}|} \pi_{\mathrm{data}}\left(\mathbf{y}_i \mid \mathbf{x}, \mathbf{y}_{<i}\right) - \prod_{i=1}^{|\mathbf{y}|} \pi_\theta\left(\mathbf{y}_i \mid \mathbf{x}, \mathbf{y}_{<i}\right)\right| \quad (16)$$

$$= \frac{1}{2}\sum_{\mathbf{y}_1, \cdots, \mathbf{y}_{|\mathbf{y}|}}\left|\prod_{i=1}^{|\mathbf{y}|} \pi_{\mathrm{data}}^{<i}(\mathbf{y}_i) - \prod_{i=1}^{|\mathbf{y}|} \pi_\theta^{<i}(\mathbf{y}_i)\right| \quad (17)$$

$$\le \frac{1}{2}\sum_{i=1}^{|\mathbf{y}|}\sum_{\mathbf{y}_1, \cdots, \mathbf{y}_i}\prod_{j=1}^{i-1}\pi_{\mathrm{data}}^{<j}(\mathbf{y}_j)\left|\prod_{i=1}^{|\mathbf{y}|}\pi_{\mathrm{data}}^{<i}(\mathbf{y}_i) - \prod_{i=1}^{|\mathbf{y}|}\pi_\theta^{<i}(\mathbf{y}_i)\right|\sum_{\mathbf{y}_{i+1}, \cdots, \mathbf{y}_{|\mathbf{y}|}}\prod_{t=i+1}^{|\mathbf{y}|}\pi_\theta^{<t}(\mathbf{y}_t) \quad (18)$$

$$= \frac{1}{2}\sum_{i=1}^{|\mathbf{y}|}\sum_{\mathbf{y}_1, \cdots, \mathbf{y}_i}\prod_{j=1}^{i-1}\pi_{\mathrm{data}}^{<j}(\mathbf{y}_j)\left|\pi_{\mathrm{data}}^{<i}(\mathbf{y}_i) - \pi_\theta^{<i}(\mathbf{y}_i)\right| \quad (19)$$

$$= \frac{1}{2}\sum_{i=1}^{|\mathbf{y}|}\sum_{\mathbf{y}_i}\mathbb{E}_{\mathbf{y}_{<i} \sim \pi_{\mathrm{data}}}\left[\left|\pi_{\mathrm{data}}^{<i}(\mathbf{y}_i) - \pi_\theta^{<i}(\mathbf{y}_i)\right|\right] \quad (20)$$

$$= \mathbb{E}_{\mathbf{y} \sim \pi_{\mathrm{data}}}\left[\sum_{i=1}^{|\mathbf{y}|}\mathrm{TV}(\pi_{\mathrm{data}}^{<i}(\mathbf{y}_i), \pi_\theta^{<i}(\mathbf{y}_i))\right], \quad (21)$$

where Equation (16) breaks the sequence-level summation into the steps and Equation (18) applies the following triangle inequality (Wen et al., 2023; Ji et al., 2023):

$$\left|\prod_{t=1}^{T} a_t - \prod_{t=1}^{T} b_t\right| \le \sum_{t=1}^{T}|a_t - b_t| \cdot \left(\prod_{i=1}^{t-1} a_i\right) \cdot \left(\prod_{j=t+1}^{T} b_j\right), \quad (22)$$

and Equation (19) marginalizes out variables $\mathbf{y}_{i+1}, \cdots, \mathbf{y}_{|\mathbf{y}|}$. □

**Theorem 3.1.** *Minimizing SFT with respect to $\theta$ is approximately minimizing the KLD between $\pi_\theta$ and the distribution of the chosen response in the preference dataset, while minimizing our SimPER is approximately minimizing the TVD.*

$$\min_\theta \mathcal{L}_{\mathrm{SFT}} \Rightarrow \min_\theta \mathrm{KL}(\pi_{\mathrm{chosen}}(\mathbf{y} \mid \mathbf{x}) \| \pi_\theta(\mathbf{y} \mid \mathbf{x})) = \sum_{y \in \mathcal{Y}} \pi_{\mathrm{chosen}}(\mathbf{y} \mid \mathbf{x}) \log \frac{\pi_{\mathrm{chosen}}(\mathbf{y} \mid \mathbf{x})}{\pi_\theta(\mathbf{y} \mid \mathbf{x})} \quad (23)$$

$$\min_\theta \mathcal{L}_{\mathrm{SimPER}} \Rightarrow \min_\theta \mathrm{TV}(\pi_{\mathrm{chosen}}(\mathbf{y} \mid \mathbf{x}) \| \pi_\theta(\mathbf{y} \mid \mathbf{x})) = \sum_{y \in \mathcal{Y}}\left|\pi_{\mathrm{chosen}}(\mathbf{y} \mid \mathbf{x}) - \pi_\theta(\mathbf{y} \mid \mathbf{x})\right| \quad (24)$$

*Proof.* Given that $\mathbb{E}_{y \sim \pi_{\text{data}}^{<i}(\mathbf{y}_i)}[\mathbf{y}_i] = \pi_{\text{data}}^{<i}(\mathbf{y}_i)$, where $\mathbf{y}_i$ represents the observed one-hot distribution with only the $w$-th index of the observed token being 1 and the others being 0, we have

$$\text{TV}(\pi_{\text{data}}^{<i}(\mathbf{y}_i), \pi_\theta^{<i}(\mathbf{y}_i)) = \frac{1}{2} \sum_{\mathbf{y}_i} |\pi_{\text{data}}^{<i}(\mathbf{y}_i) - \pi_\theta^{<i}(\mathbf{y}_i)| \tag{25}$$

$$= \frac{1}{2} \sum_{\mathbf{y}_i} |\mathbb{E}_{y \sim \pi_{\text{data}}^{<i}(\mathbf{y}_i)}[\mathbf{y}_i] - \pi_\theta^{<i}(\mathbf{y}_i)| \tag{26}$$

$$\leq \frac{1}{2} \mathbb{E}_{y \sim \pi_{\text{data}}^{<i}(\mathbf{y}_i)} \left[ \sum_{\mathbf{y}_i} |\mathbf{y}_i - \pi_\theta^{<i}(\mathbf{y}_i)| \right] = \mathbb{E}_{y \sim \pi_{\text{data}}^{<i}(\mathbf{y}_i)} \left[ \text{TV}(\mathbf{y}_i, \pi_\theta^{<i}(\mathbf{y}_i)) \right] \tag{27}$$

$$= \mathbb{E}_{y \sim \pi_{\text{data}}^{<i}(\mathbf{y}_i)} \left[ 1 - \sum_{\mathbf{y}_i} \min(\mathbf{y}_i, \pi_\theta^{<i}(\mathbf{y}_i)) \right] = -\pi_\theta^{<i}(\mathbf{y}_i). \tag{28}$$

Combing the above with Lemma A.1, we have:

$$\text{TV}(\pi_{\text{data}}(\mathbf{y} \mid \mathbf{x}), \pi_\theta(\mathbf{y} \mid \mathbf{x})) \leq \mathbb{E}_{\mathbf{y} \sim \pi_{\text{data}}} \left[ \sum_{i=1}^{|\mathbf{y}|} \text{TV}(\pi_{\text{data}}^{<i}(\mathbf{y}_i), \pi_\theta^{<i}(\mathbf{y}_i)) \right] \tag{29}$$

$$= -\mathbb{E}_{\mathbf{y} \sim \pi_{\text{data}}} \left[ \sum_{i=1}^{|\mathbf{y}|} \pi_\theta \left( \mathbf{y}_i \mid \mathbf{x}, \mathbf{y}_{<i} \right) \right]. \tag{30}$$

Recall that the objective of `SimPER`, without length-averaging, is:

$$\mathcal{L}_{\text{SimPER}}(\theta; \mathcal{D}) = -\exp\left(\log \pi_\theta(\mathbf{y}_w \mid \mathbf{x})\right) + \exp\left(\log \pi_\theta(\mathbf{y}_l \mid \mathbf{x})\right) \tag{31}$$

$$\geq -\pi_\theta(\mathbf{y}_w \mid \mathbf{x}) \geq \text{TV}(\pi_{\text{chosen}}(\mathbf{y} \mid \mathbf{x}), \pi_\theta(\mathbf{y} \mid \mathbf{x})). \tag{32}$$

Combining Equation (32) completes the whole proof. □

# B  EXPERIMENTAL DETAILS

## B.1  DOWNSTREAM TASK EVALUATION

This section is for the detailed introduction to the evaluation of downstream tasks. For models fine-tuned on UltraFeedback Binarized, following previous works (Rafailov et al., 2024; Tunstall et al., 2023), we evaluated them on HuggingFace Open LLM Leaderboard v1[1] and v2[2] (Beeching et al., 2023b; Fourrier et al., 2024), encompassing ten tasks: MMUL-PRO (Wang et al., 2024), IFEval (Zhou et al., 2023), BBH (Suzgun et al., 2022), GPQA (Rein et al., 2023), MUSR (Sprague et al., 2024), MATH (Hendrycks et al., 2021), GSM8k (Cobbe et al., 2021), ARC (Clark et al., 2018), TruthfulQA (Lin et al., 2022) and Winogrande (Sakaguchi et al., 2021) provided by the Language Model Evaluation Harness library (Gao et al., 2023b). In addition to the leaderboard, we also evaluated them on two instruction-following benchmarks AlpacaEval2 (Li et al., 2023b) and MT-Bench (Zheng et al., 2023). For models fine-tuned on Anthropic-HH for safety alignment, following previous works (Rafailov et al., 2024), we use GPT-4 for zero-shot pair-wise evaluation, which is shown to be consistent with human judgments.

**Safety Evaluation**: To evaluate the safety of the Anthropic HH trained model, the Anthropic HH test set is used as the evaluation dataset. The preferred responses within this test set serve as the baseline. GPT-4 is then employed to determine the winner, which is shown to be consistent with human judgments. We use `gpt-4-0314` for all our experiments. The specific prompts used for the GPT-4 evaluation are detailed below.

## B.2  BASELINE METHODS

This section provides a detailed introduction to state-of-the-art baselines for preference fine-tuning, with an emphasis on the usage of hyperparameters in their loss functions.

---

[1] https://huggingface.co/spaces/open-llm-leaderboard-old/open_llm_leaderboard
[2] https://huggingface.co/spaces/open-llm-leaderboard/open_llm_leaderboard

```
For the following dialogue history to a chatbot, which response is more helpful
and harmless?

Dialogue history:
{dialogue history}

Response 1:
{Response 1}

Response 2:
{Response 2}

FIRST provide a one-sentence comparison of the two responses and explain which you
feel is more helpful and harmless.  SECOND, on a new line, state only "1" or "2"
to indicate which response is more helpful and harmless.  Your response should use
the format:
Comparison:  <one-sentence comparison and explanation>
More helpful:  <"1" or "2">
```

Table 5: Prompt for GPT-4 evaluation for the safety alignment task on the Anthropic-HH dataset. {dialogue history}, {Response 1} and {Response 2} are placeholders.

**DPO** Direct Preference Optimization (Rafailov et al., 2024) uses log-likelihood differences to implicitly represent the reward function, eliminating the need for explicit reward model like RLHF. DPO involves one tunable hyperparameter, $\beta$, which controls the deviation from the reference model.

$$\mathcal{L}_{\text{DPO}}(\theta; \mathcal{D}) = \mathbb{E}_{(\mathbf{x}, \mathbf{y}_w, \mathbf{y}_l) \sim \mathcal{D}} \left[ -\log \sigma(\beta \log \frac{\pi(\mathbf{y}_w \mid \mathbf{x})}{\pi_{\text{ref}}(\mathbf{y}_w \mid \mathbf{x})} - \beta \log \frac{\pi(\mathbf{y}_l \mid \mathbf{x})}{\pi_{\text{ref}}(\mathbf{y}_l \mid \mathbf{x})}) \right].$$

**IPO** Identity Preference Optimization (Azar et al., 2024) minimizes a squared loss regression problem by defining an alternative reward function, avoiding unstable RL training. IPO involves one hyperparameter, $\beta$, to adjust the reward margin.

$$\mathcal{L}_{\text{IPO}}(\theta; \mathcal{D}) = \mathbb{E}_{(\mathbf{x}, \mathbf{y}_w, \mathbf{y}_l) \sim \mathcal{D}} \left[ \left( \log \frac{\pi_\theta(\mathbf{y}_w \mid \mathbf{x}) \pi_{\text{ref}}(\mathbf{y}_l \mid \mathbf{x})}{\pi_\theta(\mathbf{y}_l \mid \mathbf{x}) \pi_{\text{ref}}(\mathbf{y}_w \mid \mathbf{x})} - \frac{1}{2\beta} \right)^2 \right].$$

**CPO** Contrastive Preference Optimization (Xu et al., 2024) uses log-likelihood as the reward and is trained alongside a Supervised Fine-Tuning (SFT) objective. CPO involves two hyperparameters: $\beta$, which scales the log probabilities, and $\lambda$, which weights the SFT component.

$$\mathcal{L}_{\text{CPO}}(\theta; \mathcal{D}) = -\log \sigma \left( \beta \log \pi_\theta(\mathbf{y}_w \mid \mathbf{x}) - \beta \log \pi_\theta(\mathbf{y}_l \mid \mathbf{x}) \right) - \lambda \log \pi_\theta(\mathbf{y}_w \mid \mathbf{x}).$$

**SLiC** Sequence Likelihood Calibration (Zhao et al., 2023) directly uses log-likelihood and includes a SFT objective. SLiC involves two hyperparameters: $\delta$, which sets the margin for the ranking loss, and $\lambda$, which weights the SFT component.

$$\mathcal{L}_{\text{SLiC}}(\theta; \mathcal{D}) = \max \left( 0, \delta - \log \pi_\theta(\mathbf{y}_w \mid \mathbf{x}) + \log \pi_\theta(\mathbf{y}_l \mid \mathbf{x}) \right) - \lambda \log \pi_\theta(\mathbf{y}_w \mid \mathbf{x}).$$

**SimPO** Simple Preference Optimization (Meng et al., 2024) eliminates the need for a reference model and optimizes a length-regularized probability of response pairs. SimPO involves two hyperparameters: $\beta$ to scale the log probabilities and $\gamma$ to adjust the reward margin.

$$\mathcal{L}_{\text{SimPO}}(\theta; \mathcal{D}) = \mathbb{E}_{(\mathbf{x}, \mathbf{y}_w, \mathbf{y}_l) \sim \mathcal{D}} \left[ -\log \sigma(\beta \log \pi_\theta(\mathbf{y}_w \mid \mathbf{x}) - \beta \log \pi_\theta(\mathbf{y}_l \mid \mathbf{x}) - \gamma) \right].$$

**KTO** Kahneman-Tversky Optimization (Ethayarajh et al., 2024b) learns from non-paired preference data. KTO involves three hyperparameters: $\beta$, which controls the deviation from the reference model; $\lambda_w$ and $\lambda_l$, which weight the preference components for winning and losing responses, respectively.

$$\mathcal{L}_{\text{KTO}}(\theta; \mathcal{D}) = -\lambda_w \sigma \left( \beta \log \frac{\pi_\theta(\mathbf{y}_w \mid \mathbf{x})}{\pi_{\text{ref}}(\mathbf{y}_w \mid \mathbf{x})} - z_{\text{ref}} \right) + \lambda_l \sigma \left( z_{\text{ref}} - \beta \log \frac{\pi_\theta(\mathbf{y}_l \mid \mathbf{x})}{\pi_{\text{ref}}(\mathbf{y}_l \mid \mathbf{x})} \right), \tag{33}$$

$$\text{where} \quad z_{\text{ref}} = \mathbb{E}_{(x,y) \sim \mathcal{D}} \left[ \beta \, \text{KL} \left( \pi_\theta(\mathbf{y} \mid \mathbf{x}) || \pi_{\text{ref}}(\mathbf{y} \mid \mathbf{x}) \right) \right].$$

B.3 IMPLEMENTATION DETAILS

**Training Hyperparameters.** For general hyperparameters, we adhered strictly to the settings used in SimPO. We applied the following hyperparameters: For the SFT stage, we use a learning rate of $2 \times 10^{-5}$. For both the SFT and the preference optimization stages, we use a batch size of 128, a maximum sequence length of 2048, and a cosine learning rate schedule with 10% warmup steps for one epoch, all through the Adam optimizer (Kingma, 2014). We maintain these settings consistently to ensure uniformity and comparability across experiments.

For method-specific hyperparameters, we also followed the search strategy from SimPO, noting that our method does not require any additional hyperparameters beyond the learning rate. Specifically, for different baseline methods, each with its own set of additional hyperparameters, the search strategy is in Table 6. Each method is individually search for the learning rates in $[3e-7, 5e-7, 6e-7, 1e-6]$.

Table 6: Search ranges for hyperparameters across baseline methods

| Method | Hyperparameter Search Range |
|---|---|
| DPO | $\beta \in [0.01, 0.05, 0.1]$ |
| IPO | $\tau \in [0.01, 0.1, 0.5, 1.0]$ |
| CPO | $\lambda = 1.0$ 
 $\beta \in [0.01, 0.05, 0.1]$ |
| SLiC | $\lambda \in [0.1, 0.5, 1.0, 10.0]$ 
 $\delta \in [0.1, 0.5, 1.0, 2.0]$ |
| KTO | $\lambda_l = \lambda_w = 1.0$ 
 $\beta \in [0.01, 0.05, 0.1]$ |
| SimPO | $\beta \in [2.0, 2.5]$ 
 $\gamma \in [0.3, 0.5, 1.0, 1.2, 1.4, 1.6]$ |

**Evaluation Hyperparameters.** The hyperparameters for evaluation on AlpacaEval 2 adhere to SimPO, with a temperature of 0.7 for the Mistral-Base setting, a temperature of 0.5 for the Mistral-Instruct setting, and a temperature of 0.9 for both Llama3 settings. The hyperparameters for evaluation on MT-Bench are set according to the official decoding configuration, which specifies distinct sampling temperatures for various categories.

**Computation Resources.** All the training experiments in this paper were conducted on 4 × NVIDIA A100 (80G) GPUs with 128 batchsize, based on the alignment-handbook repo.[3]

## C CASE STUDY

In this section, we present examples of comparisons between `SimPER` and the baseline `SimPO`.

---

[3] https://github.com/huggingface/alignment-handbook

| Prompt | Does the 5 second rule actually exist? |
|---|---|
| SimPER | The "5 second rule" is an urban legend that claims that if food falls on the floor, it is still safe to eat as long as you pick it up within 5 seconds. However, this rule is not a scientifically proven fact, and there is no evidence to support it. Here are some reasons why the 5-second rule is not a reliable method for determining food safety:

1. **Bacteria and viruses can transfer quickly:** Bacteria and viruses can transfer from surfaces to food within seconds. The amount of time that passes between a food item's contact with a contaminated surface and its being picked up can affect the likelihood of bacterial or viral transmission.

2. **Contamination levels on floors and surfaces:** The level of contamination on floors and surfaces can vary widely depending on the environment, the cleanliness of the area, and the presence of moisture. Even if a surface is dry, bacteria and viruses can still be present.

3. **Personal hygiene:** The cleanliness of your hands, the surface you place the food on, and the utensils you use to pick up the food all play a role in food safety. Picking up food within 5 seconds does not ensure that it is clean enough to eat.

4. **Temperature:** The temperature of food can also affect the risk of bacterial or viral contamination. Food that has been at room temperature for a prolonged time, even if picked up quickly, may still harbor dangerous bacteria.

Therefore, it is best to follow proper food handling and preparation practices, such as washing hands before and after handling food, using clean utensils and surfaces, and cooking food to the appropriate temperatures to ensure its safety. |
| SimPO | The "5-second rule" refers to a popular myth or belief that suggests touching food on the ground for less than five seconds makes it safe to consume without contamination. However, this rule does not exist scientifically. According to food safety guidelines, bacteria can transfer from contaminated surfaces to food within seconds, regardless of the elapsed time. The duration of contact between the food and contaminated surfaces plays a crucial role in determining the likelihood of bacterial transfer, but it's not limited to five seconds. Cross-contamination occurs when bacteria from unwashed hands, dirty utensils, or contaminated surfaces come into contact with food, leading to potential foodborne illnesses. Therefore, adhering to proper hygiene practices, such as washing hands, cleaning surfaces, and cooking food thoroughly, remains essential to prevent food contamination. In summary, there is no scientific basis for the "5-second rule," and relying solely on this belief could increase the risk of consuming contaminated food." |

Table 7: An AlpacaEval 2 generation from the Llama-3-8B-Base model trained with `SimPER` and `SimPO`. Alpaca-Eval-GPT4-Turbo prefers `SimPER` over `SimPO`. Compared to `SimPO`, the generation by `SimPER` is better structured, with hierarchical discussions that make it more helpful.

| | |
|---|---|
| **Prompt** | How do I make escargot? |
| **SimPER** | Here's a general recipe for making escargot: |

```
Ingredients:
- 1 pound fresh snails (escargots)
- 1/4 cup butter, at room temperature
- 1 clove garlic, minced
- 2 tablespoons finely chopped parsley
- 2 tablespoons finely chopped shallots
- 2 tablespoons dry white wine
- 2 tablespoons lemon juice
- 1/4 teaspoon ground black pepper
- 1/4 cup breadcrumbs
- 1/4 cup grated Parmesan cheese
- Salt, to taste
- 1 tablespoon olive oil

Equipment:
- Snail shells
- Foil
- A heavy-bottomed skillet
- A wooden spoon
- A baking dish
- A fine-mesh strainer
- A small saucepan
- A mixing bowl
- Parchment paper

Instructions:
1.  Preheat the oven to 375F (190C).
2.  Scrub the snails under cool running water to remove any dirt or
debris.  Discard any snails that are not alive.  Soak the snails in
salted water for at least 30 minutes, changing the water every 10
minutes, to ensure they are thoroughly cleaned.
3.  In a small saucepan, melt the butter over low heat.  Add the
minced garlic, parsley, and shallots, and cook for 1-2 minutes until
fragrant.
4.  Remove the garlic mixture from the heat and stir in the white
wine, lemon juice, and black pepper.  Season with salt to taste.
5.  In a mixing bowl, combine the breadcrumbs and grated Parmesan
cheese.
6.  Place the snail shells on a sheet of foil and arrange them in
a baking dish.  Spoon the garlic butter mixture into the shells,
filling each one about 34 full.
7.  Sprinkle the breadcrumb mixture over the top of each snail.
8.  Cover the snail shells with foil.
9.  Bake in the preheated oven for 12-15 minutes or until the
breadcrumbs are golden brown and the snails have cooked through.
10.  Remove the dish from the oven, carefully remove the foil, and
place a small piece of parchment paper over the top of each shell.
11.  Return the dish to the oven and continue baking for 1-2 minutes
to crisp up the breadcrumbs.
12.  Remove the dish from the oven, and let the escargots cool for a
few minutes.
13.  In a heavy-bottomed skillet, heat the olive oil over medium-high
heat.  Strain the garlic butter mixture through a fine-mesh strainer
into the skillet, and cook for 1-2 minutes until slightly reduced and
thickened.
14.  To serve, spoon the garlic butter onto small plates, arrange the
escargots on the plates, and drizzle the remaining garlic butter over
the top.
Note:  Escargot shells are available for purchase at specialty food
stores or online.  If you don't have escargot shells, you can use
mushroom caps or even small ramekins.
Enjoy your delicious escargot!
```

---

**SimPO** Here's a general recipe for making escargot:

```
Ingredients:
- 1 pound fresh or canned snails (escargots), cleaned and cooked
- 1/2 cup unsalted butter, softened
- 2 cloves garlic, minced
- 1 tablespoon finely chopped fresh parsley
- 1 tablespoon finely chopped fresh thyme
- 1/4 teaspoon red pepper flakes (optional)
- Salt and freshly ground black pepper, to taste
- 24-30 small, hollowed-out garlic cloves or escargot shells

Instructions:
1.  Preheat your oven to 450F (230C).
2.  Prepare the garlic cloves or escargot shells by gently scraping
out the pulp with knife, leaving a thin layer intact.  Rinse them
under running water and pat dry.
3.  In a mixing bowl, combine softened butter, minced garlic, chopped
parsley, thyme, red pepper flakes (if using), salt, and black pepper.
Mix well until blended.
4.  Spoon a generous amount of seasoned butter into each garlic clove
or escargot shell, filling them almost to the top.
5.  Arrange the filled shells in a baking dish, preferably one with
individual wells or indentations, spaced evenly apart.
6.  Place the cooked snails into each shell, making sure they fit
snugly.
7.  Bake the escargot for 8-10 minutes, or until the butter has
melted and bubbled around the edges, and the snails are heated
through.
8.  Remove the dish from the oven and serve immediately, accompanied
by crusty bread, toasted baguette slices, or crackers.
9.  To eat, use a small fork or escargot spoon to extract the snails
from their shells, dip them into the remaining garlic butter, and
enjoy!
Note:  Always follow proper food handling and preparation guidelines
when working with live or cooked snails, especially if using fresh
ones.  Ensure they are properly cleaned, cooked, and stored according
to recommended safety practices.
```

---

Table 8: An AlpacaEval 2 generation from the Mistral-7B-Base model trained with `SimPER` and `SimPO`. Alpaca-Eval-GPT4-Turbo prefers `SimPER` over `SimPO`. Compared to `SimPO`, the generation by `SimPER` is better structured, with hierarchical discussions that make it more helpful.

---

| Prompt | Mick pays his teacher \$800 for 40 lessons worth 2 hours each.  If this will be all he is going to pay for his lessons, how much did he receive? |
|---|---|
| **SimPER** | Mick did not receive any money.  The \$800 he paid to his teacher is the total cost for the 40 lessons, which is 40 x 2 hours = 80 total hours of lessons.  Since he only paid for the lessons, he will not receive any money back.
Confidence:  95% |
| **SimPO** | Mick received \$800.
Confidence:  95% |

Table 9: An AlpacaEval 2 generation from the Llama-3-8B-Base model trained with `SimPER` and `SimPO`. Alpaca-Eval-GPT4-Turbo prefers `SimPER` over `SimPO`. Compared to `SimPO`, `SimPER`'s output provides clearer step-by-step reasoning.

