# OpenReview forum: "SimPER: A Minimalist Approach to Preference  Alignment without Hyperparameters"
_ICLR.cc/2025/Conference — ICLR 2025 Poster_

### Official Review · Reviewer_MQ2x · 2024-10-29

**Soundness:** 3
**Presentation:** 3
**Contribution:** 3
**Rating:** 6
**Confidence:** 4

**Summary:**

The paper presents a novel loss SimPER for preference learning, which minimizes (resp. maximizes) the perplexity on chosen (resp. rejected) completions and does not introduce any hyper-parameter.
SimPER is applied to align the base and instruct version of Mistral-7B and Llama-3-8B on UltraFeedback and Pythia-2.8B on Anthropic Helpful and Harmless dataset.
The aligned models are evaluated on Open LLM Leaderboard and instruction-following benchmarks AlpacaEval 2 and MT-Bench, where SimPER leads to models with both superior basic capabilities and alignment.

**Strengths:**

* The paper addresses an important and notorious problem, i.e., sensitivity of alignment algorithms to hyper-parameters.
* In the empirical evaluation, the proposed method achieves significantly higher basic capabilities and alignment.

**Weaknesses:**

* If the exponential operation in eq.(8) is removed, the loss degrades into standard fine-tuning that learns the chosen completion and unlearns the rejected completion.
It will be beneficial to compare with the standard fine-tuning and elaborate why adding the exponential operation is better.
* I disagree with the analysis on line 249. The authors neglect that the term $w_\theta$ (resp. $d_\theta)$ also depends on $\pi_\theta(y_l|x)$ (resp. $p_\theta(y_l|x))$. For example, in DPO, the weight of $\nabla_\theta\pi_\theta(y_l|x)$ is
$$w=\frac{\beta\sigma(\beta\log\pi_\theta(y_l|x)+C)}{\pi_\theta(y_l|x)}$$
and we have $w\to 0$ when $\pi_\theta(y_l|x)\to 0$.
* My primary concern is whether the proposed method can be scaled to iterative preference learning [1, 2], which is currently employed by the state-of-the-art open-weight models [3, 4].
For instance, implementing iterative DPO on Llama-3-8B-Base can achieve advanced performance on academic benchmarks while scoring 31.3 on AlpacaEval 2 and 8.46 on MT-Bench [5], significantly outperforming offline setting.
Conducting such evaluation should be straight-forward given the only difference is the loss term?
* I noticed that the Table 2 is largely copied from [6], which is fine but contradicts with the statement “We thoroughly tuned the hyper parameters for each baseline” on line 371? By the way the MT-Bench score for Llama-3-8B-Base seems was mistakenly duplicated as the result where GPT-4-Turbo serves as the judge.
* Could you please provide the scores for SFT model in Tab. 3 to illustrate the alignment tax?
* Minor: typo in eq. (4) and expression for $d_\theta$ on line 244.

[1] Self-Play Preference Optimization for Language Model Alignment, arXiv.
[2] Self-Exploring Language Models, arXiv.
[3] The Llama 3 Herd of Models, arXiv.
[4] Qwen2 Technical Report, arXiv.
[5] RLHF Workflow: From Reward Modeling to Online RLHF, arXiv.
[6] SimPO: Simple Preference Optimization with a Reference-Free Reward, NeurIPS'24.

**Questions:**

- Given that the loss can be decomposed into the terms corresponding to chosen and rejected completions (eq. 7), does SimPER not require the chosen and rejected completions appear in pairs like KTO, making the algorithm more general?

---

> ### Author Response · Authors · 2024-11-20
> **Response to Reviewer MQ2x Part (1/2)**
>
> Dear reviewer MQ2x, we appreciate your time, comments, and suggestions very much! However, we believe that there may have been some misunderstandings. Therefore, we would like to provide a point-by-point response to your comments.
>
> ---
> **Q1.  If the exponential operation in eq.(8) is removed, the loss degrades into standard fine-tuning that learns the chosen completion and unlearns the rejected completion. It will be beneficial to compare with the standard fine-tuning and elaborate why adding the exponential operation is better?**
>
> **A1**. Thank you for your valuable suggestion! We conducted an ablation study by removing the exponential operation, and the results are provided in Table 5 in the updated submission.
>
> We observe that the performance becomes significantly worse than the baselines when the exponential operation is removed. The reason is that the norm of the gradient on rejected responses becomes very large without the exponential operation, as the gradient of the model probability is weighted by the reciprocal of the model probability for the rejected response.
>
> We also provide the training dynamics of removing the exponential operation in Figure 7 in the updated submission, which further verifies that the likelihood of both rejected and chosen responses decreases rapidly when the exponential operation is removed.
>
>
>
> ---
>
> **Q2.  I disagree with the analysis on line 249. The authors neglect the weight term $w_{\theta}$.**
>
> **A2.** **We apologize for any confusion and believe there may have been some misunderstandings. While there is a weight term applied to the gradients of rejected responses, it is also shared by chosen responses. Our analysis specifically focuses on the gradient ratio between chosen and rejected responses.** Specifically, the gradient ratio between the decrease in the probability of rejected responses and the increase in the probability of chosen responses is:
> $$
> \frac{\pi_{\theta}(y_{w} \mid x)}{\pi_{\theta}(y_{l} \mid x)} \cdot \frac{\nabla_{\theta} \pi_{\theta}(y_{l} \mid x)}{\nabla_{\theta} \pi_{\theta}(y_{w} \mid x)},
> $$
> which becomes infinite when $\pi_{\theta}(y_{l} \mid x) \rightarrow 0$. A larger gradient ratio leads to a faster decrease in the probability of a rejected response compared to the increase in the probability of a chosen response during the training process (i.e., the model emphasizes rejected responses more than chosen responses). In contrast, our SimPER alleviates this issue because the gradient of the model probability is not weighted by the reciprocal of the model probability. We apologize for any confusion caused and have clarified this point more explicitly in the updated submission (marked with red text).
>
>
> ---
>
> **Q3. My primary concern is whether the proposed method can be scaled to iterative preference learning, which is currently employed by the state-of-the-art open-weight models.**
>
> **A3.** Thank you for your comments.  **Our works focus exclusively on offline settings, avoiding any iterative training processes. Nevertheless, we have discussed and cited the papers mentioned by the reviewers  in Section 4.2 and completely agree that including an empirical comparison on iterative training processes would be valuable.**
>
> Following your suggestion, we are actively working on an iterative version of SimPER. Specifically, we alternate between generating online preference optimization data and training the Mistral-7B-Instruct model with labeled preference data using the PairRM reward model. From the following results, we find that the online version of SimPER generally continues to improve performance with each additional iteration. These results have also been included in the updated submission .
>
> | Methods |  AlpacaEval2 LC  | AlpacaEval2 WR | MT-Bench  | GPQA | MATH |TruthfulQA |
> | --- | --- | --- | --- | --- |--- |--- |
> | SimPO   | &nbsp; &nbsp; 32.1|  &nbsp; &nbsp; 34.8   |  7.6 | 28.04 | 2.19  | 68.40 |
> | SimPER (1 Iter)  | &nbsp; &nbsp; 37.8  | &nbsp; &nbsp; 39.5  |  7.8 | 28.36 | 2.51 | 68.78 |
> | SimPER  (2 Iter)    | &nbsp; &nbsp; 38.5  | &nbsp; &nbsp;  41.1  |  **7.9** | 28.52 | 2.62 | **68.92** |
> | SimPER  (3 Iter)   | &nbsp; &nbsp; **40.2** |  &nbsp; &nbsp; **43.5**  | **7.9** | **29.37** | **2.65**  | 68.41 |
> ---
>
> **Q4. I noticed that Table 2 is largely copied from SimPO [6], which is fine but contradicts with the statement “We thoroughly tuned the hyper parameters for each baseline” on line 371?**
>
> **A4.** Thank you for your comment. Yes, for Table 2, the baseline results are taken from SimPO, as we found the code provided by SimPO to be fully reproducible, and we strictly followed the same settings as in SimPO. The statement 'We thoroughly tuned the hyperparameters for each baseline' refers specifically to the safety alignment task. We apologize for any confusion caused and have clarified this point in the updated submission.

---

> ### Author Response · Authors · 2024-11-20
> **Response to Reviewer MQ2x Part (2/2)**
>
> **Q5. By the way the MT-Bench score for Llama-3-8B-Base seems was mistakenly duplicated as the result where GPT-4-Turbo serves as the judge.**
>
> **A5.** Thank you for your comments, and we apologize for any confusion. We utilized GPT-4, not GPT-4-Turbo, as the judge. However, there was a single typo: SFT should be 6.6 instead of 5.2. We have corrected this.
>
>
> ---
>
> **Q6. Could you please provide the scores for the SFT model in Tab. 3 to illustrate the alignment tax?**
>
> **A6.** Thank you for your suggestions. **We have updated Table 3 to include the results of the SFT model. Please refer to the updated submission PDF.** The results demonstrate that SimPER can more effectively preserve reasoning abilities than SimPO and DPO, such as math and abstract reasoning, and even significantly outperforms SFT in many cases. Consequently, SimPER incurs less alignment tax given its superior performance on alignment benchmarks.
>
> ---
>
>
> **Q7. Minor: typo in eq. (4) and expression for  $d_{\theta}$  on line 244.**
>
> **A7.** Thank you for pointing out the typos! We have corrected them.
>
> ---

---

> ### Author Response · Authors · 2024-11-20
> **Summary of Response to Reviewer MQ2x**
>
> Dear reviewer MQ2x, we appreciate your comments and sincerely thank you for your feedback. We have made our utmost effort to address your comments in the rebuttal. **We clarified several details to address your comments, suggestions and misunderstandings .
> We also further evaluated SimPER in the online iterative scenario.**
>
>
> **As all reviewers have noted, SimPER presents a simple yet effective approach, offering valuable insights and addressing a very important and practical real-world problem i.e., sensitivity of alignment objectives to hyper-parameters. As your comments are indeed not fatal to the major contributions of our manuscript, could you please consider raising your score in light of our efforts? Many thanks for your time; we are extremely grateful!**

---

> > ### Comment · Reviewer_MQ2x · 2024-11-20
> >
> > I thank the authors for their response.
> > My concerns are resolved so I have raised my score.

---

> > > ### Author Response · Authors · 2024-11-20
> > > **Author Response to Reviewer MQ2x**
> > >
> > > Dear Reviewer MQ2x,
> > >
> > > We are very glad to hear that our rebuttal have adequately addressed your concerns.
> > >
> > > We sincerely appreciate your time and effort.
> > >
> > > Best regards,
> > >
> > > Authors

---

### Official Review · Reviewer_dfTo · 2024-10-30

**Soundness:** 3
**Presentation:** 3
**Contribution:** 3
**Rating:** 6
**Confidence:** 4

**Summary:**

This paper proposes a simple offline preference optimization objective that eliminates the need for a reference model and any tunable hyperparameters required by previous methods, enhancing LLM alignment performance.

**Strengths:**

1. The issue of reduced log probability for the chosen response has recently gained considerable attention in the field of LLM alignment. Addressing this issue is a valuable contribution.
2. SimPER reduces the decrease in the likelihood of chosen responses, providing an improvement over SimPO.
3. The experiments are thorough and well-executed, with comprehensive evaluations on both the Open LLM Leaderboard and various chat benchmarks, yielding promising results.

**Weaknesses:**

1. It appears that Equation (7) might lack an expectation over $x$. If an expectation with respect to $x$ is included, then based on the property $E(A + B) = E(A) + E(B)$, would SimPER’s application extend beyond the paired preference data scenario described in the paper? This could incorporate comparisons with unpaired alignment algorithms.

2. While Figure 3 shows that SimPER mitigates the reduction in chosen response log probabilities compared to SimPO, the likelihood of the chosen response is still decreasing. The ORPO paper addresses this issue as well, and Figure 7 in ORPO shows that the likelihood of the chosen response can actually increase.

3. On Line 295, "model covering" should likely be "model seeking”?

**Questions:**

See weakness section.

---

> ### Author Response · Authors · 2024-11-20
> **Response to Reviewer dfTo**
>
> Dear reviewer dfTo, we sincerely appreciate your recognition of our contributions and your positive comments on our work, such as 'well-written,' 'well-motivated,' 'simple with significant advantages,' and 'sound experiments.' Please find our responses to your comments below:
>
> ---
> **Q1. Would SimPER’s application extend beyond the paired preference data scenario described in the paper? This could incorporate comparisons with unpaired alignment algorithms.**
>
> **A1.** Thank you for your question! SimPER can be extended to unpaired data, as it only requires either single chosen or rejected samples without the need for pairwise data, similar to KTO. To test this, we randomly discarded the chosen responses or rejected responses in increasingly large fractions of  pairwise data while aligning the Llama3-8B-Base model. Please refer to Table 8 in the updated submission. These additional results demonstrate that even without pairwise data (e.g., discarding up to 50\% of the chosen or rejected responses), SimPER still outperforms SimPO and performs better than the baseline KTO, which also does not require pairwise data.
>
>
> | Llama3-8B-Base |  AlpacaEval2 LC   | AlpacaEval2 WR  | ARC | TruthfulQA| GSM8K |
> | --- | --- | --- | --- | --- |--- |
> | SimPO  | 22.0  | 20.3  |  65.19 | 59.46 |  31.54  |
> | KTO |  14.2    |  12.4   |  63.14  | 55.76 | 38.97 |
> | KTO (50 %)  |  12.7 |  10.5  | 65.52 | 58.73 | 45.31 |
> | SimPER  | 25.2  |  22.9   |  67.06  |  62.59 | 51.02  |
> | SimPER (50 %)  | 23.5  | 21.3  |  68.31 | 63.25 | 49.16 |
>
>
> ---
> **Q2. While Figure 3 shows that SimPER mitigates the reduction in chosen response log probabilities compared to SimPO, the likelihood of the chosen response is still decreasing. The ORPO paper addresses this issue as well, and Figure 7 in ORPO shows that the likelihood of the chosen response can actually increase.**
>
> **A2.** Thank you for your comments! While ORPO can significantly increase the likelihood of the chosen response, it performs worse than SimPER on many benchmarks (please refer to the additional results of ORPO in Table 5 of the updated submission).
>
> One possible reason is that the preference datasets are not always of high quality. Often, the chosen response is only slightly better than the rejected one. In such cases, it is not necessarily desirable for the model to significantly increase the probability mass on the chosen response (i.e., overfit to the chosen response). In contrast, SimPER prevents a decrease in the likelihood of the chosen response while also avoiding overfitting to it through a hyperparameter-free objective
>
>
>
>
> ---
> **Q3. On Line 295, "model covering" should likely be "model seeking”?**
>
> **A3.** Thanks for pointing out these typos! We have corrected them.
>
>
> ---
>
> **We appreciate the efforts from the reviewer and also sincerely hope our posted responses can address your questions. We also believe your comments can also be easily addressed in the revision. In light of these responses, we sincerely hope you could consider increasing your score. Many thanks for your time; we are extremely grateful!**

---

> > ### Comment · Reviewer_dfTo · 2024-11-25
> >
> > Thank you for your response! I keep my positive score.

---

> ### Author Response · Authors · 2024-11-25
> **Response to Reviewer dfTo**
>
> We sincerely appreciate your positive feedback and review. Thank you very much for your time and reply.
>
> Best wishes,
>
> Authors

---

### Official Review · Reviewer_bGmT · 2024-10-31

**Soundness:** 3
**Presentation:** 4
**Contribution:** 4
**Rating:** 8
**Confidence:** 3

**Summary:**

The authors introduced SimPER, a preference optimization technique based on inverse perplexity that eliminates the need for a reference model and hyperparameters. This makes it straightforward to implement and computationally efficient. They demonstrated that SimPER effectively minimizes the Total Variation Distance, which promotes the generation of a subset of high-reward responses (mode-covering behavior) rather than distributing probability equally across all responses. The authors showed that SimPER compares favorably against DPO, IPO, KTO, and SimPO when tested on both pre-trained and instruction-tuned Mistral-7B and LLaMA3-8B models.

**Strengths:**

- Preference alignment is a timely and impactful area of research.
- The proposed method, SimPER, is well-motivated. It eliminates the need for a reference model and hyperparameters, making it simple to implement and cost-effective, which are significant advantages.
- The experimental setup is sound and modern, incorporating popular and recent LLMs and benchmarks.
- The paper is well-written and easy to follow.

**Weaknesses:**

- The authors did not compare SimPER with ORPO [1], another reference model-free preference alignment method.
- Some necessary details and ablations are missing (see Questions).

**Questions:**

- Can you include ORPO [1] in the experimental setup?
- How sensitive are other preference alignment methods to hyperparameters? Are instruction-tuned models consistently less sensitive to the hyperparameters of the preference alignment methods? (Figure 1)
- What is the training dynamic of DPO mentioned in line 253? (Figure 3)
- How was the reference model incorporated in SimPER in the ablation study? (Table 4)
- Line 72, "post-training process for alignment is usually very expensive", I suggest mentioning the LoRA line of work, which made it significantly less expensive [2, 3].
- Line 201, "data. (Jelinek et al., 1977; Marion et al., 2023; Gonen et al., 2023).", there is a period before the citation.
- Figure 6, I suggest decreasing the opacity as it is difficult to read.

[1] Hong, J., Lee, N., & Thorne, J. (2024). Reference-free monolithic preference optimization with odds ratio. arXiv preprint arXiv:2403.07691.
[2] Hu, E. J., Shen, Y., Wallis, P., Allen-Zhu, Z., Li, Y., Wang, S., ... & Chen, W. (2021). Lora: Low-rank adaptation of large language models. arXiv preprint arXiv:2106.09685.
[3] Dettmers, T., Pagnoni, A., Holtzman, A., & Zettlemoyer, L. (2024). Qlora: Efficient finetuning of quantized llms. Advances in Neural Information Processing Systems, 36.

---

> ### Author Response · Authors · 2024-11-20
> **Response to Reviewer bGmT**
>
> Dear reviewer bGmT, we appreciate your positive feedback on our paper's motivation, soundness, novel insights, and contribution. Please find our detailed responses below:
>
> ---
> **Q1. The authors did not compare SimPER with ORPO [1], another reference model-free preference alignment method. Can you include ORPO [1] in the experimental setup**
>
>
> **A1.** Thank you for your suggestions! We further compared SimPER with ORPO across multiple benchmarks, with the results provided in Table 5 of the updated submission. These additional results show that SimPER significantly outperforms ORPO. This is particularly encouraging as SimPER not only operates as a reference model-free preference optimization method but is also hyperparameter-free, unlike ORPO, further highlighting the strength of our contribution.
>
>
> ---
> **Q2. How sensitive are other preference alignment methods to hyperparameters? Are instruction-tuned models consistently less sensitive to the hyperparameters of the preference alignment methods? (Figure 1).**
>
> **A2.** Thank you for your questions. We provided additional results for DPO, KTO, and IPO with different hyperparameters on the Mistral-Base and Mistral-Instruct models in Figure 8 of the updated submission. We observe that these methods are also especially sensitive to hyperparameters, similar to SimPO in Figure 1. Moreover, we find that instruction-tuned models are slightly more robust to the hyperparameters of preference alignment methods. One possible reason may be that our Instruct setup is closer to an on-policy setting, resulting in a smaller distribution shift, which leads to greater robustness.
>
>
> ---
> **Q3. What is the training dynamic of DPO mentioned in line 253? (Figure 2)**
>
>
> **A3.** Thank you for your question. We also provided the training dynamics of DPO in Figure 7 in the updated submission. We can observe that the likelihood of both chosen and rejected responses continues to decrease, which further supports our discussion in Section 3.3.
>
>
> ---
> **Q4. How was the reference model incorporated in SimPER in the ablation study?**
>
> **A4.** Thank you for your question. The objective that incorporates reference model in SimPER is
> $$-\exp\left(\frac{1}{|y_w|} \log \frac{\pi_\theta(y_w \mid x)}{\pi_{\text{ref}}(y_w \mid x)}\right)+\exp\left(\frac{1}{|y_l|} \log \frac{\pi_{\theta}(y_l \mid x)}{\pi_{\text{ref}}(y_l \mid x)}\right).$$
> We will clearly state this in the main text.
>
>
>
>
> **Q5. Line 72, "post-training process for alignment is usually very expensive", I suggest mentioning the LoRA line of work, which made it significantly less expensive [2, 3].**
>
> **A5.** Thank you for your suggestion. We have cited these works and added a discussion in the related work section of our updated submission.
>
> **Q6. Line 201, "data. (Jelinek et al., 1977; Marion et al., 2023; Gonen et al., 2023).", there is a period before the citation.
> Figure 6, I suggest decreasing the opacity as it is difficult to read.**
>
> **A6.** Thank you for your comments! We have corrected the typo and updated Figure 6 by decreasing the opacity.
>
> **We thank the reviewer for providing valuable comments on our paper. In light of these responses, we hope we have addressed your questions and sincerely hope you consider raising your score. Many thanks for your time; we are extremely grateful!**

---

> > ### Comment · Reviewer_bGmT · 2024-11-22
> >
> > Thank you for the response. The authors have addressed all of my concerns and questions, and I am pleased with the clarifications provided. I will be maintaining my score and hope that your submission is accepted. Wishing you the best with your paper.

---

> > > ### Author Response · Authors · 2024-11-22
> > > **Author Response to Reviewer bGmT**
> > >
> > > Dear Reviewer bGmT,
> > >
> > > Thank you very much for reviewing our paper and reading our rebuttal. We sincerely appreciate your recognition of our contribution.
> > >
> > > We are truly grateful for your time and your reply.
> > >
> > > Authors

---

### Official Review · Reviewer_7YzN · 2024-11-02

**Soundness:** 3
**Presentation:** 3
**Contribution:** 4
**Rating:** 6
**Confidence:** 3

**Summary:**

Existing preference optimization techniques often require complex hyperparameter tuning, which complicates the fine-tuning process. In this work, the authors propose a new method for aligning large language models (LLMs) with human preferences. To address this, they propose SimPER, a hyperparameter-free approach that optimizes inverse perplexity—defined as the inverse of the exponentiated average log-likelihood of chosen and rejected responses. They then provide some insights on the reason that the method can work better than other widely adopted methods that SimPER does not push the model away from the chosen responses. They theoretically show that minimizing the proposed loss is approximately minimizing TVD, which is more immune to mode dropping. They validate the method by many commonly adopted benchmarks, and show the method have superior performance without additional hyperparameters.

**Strengths:**

Hyperparameter tuning has long been a headache in posttraining. SimPER consistently outperforms its alternatives across multiple tasks and datasets without meticulous adjustment of various parameters. This not only is efficient in training, but also enhances the robustness of the model.

In addition to the empirical performance, the authors also provide insights on why such a loss can perform better than its alternatives.

The proposed loss is fairly easy to implement.

The proposed method has less length bias.

**Weaknesses:**

It would be better if the experiments can show the proposed method works well across different scale of models. The majority of the experiments adopts 7-8B models. It would be ideal if we can show usefulness with, e.g., ~3B, ~7B, ~70B models.

In addition, I am confused on the choice of the model. It seems only Fig.4 uses Pythia-2.8B, but all other experiments are with Mistral-7B and/or Llama3-8B? Can you explain why this is the case?

The proposed loss is fairly simple and I do believe simplicity is good. But the presentation of Eq. (5)(6)(7)(8) is redundant. Assuming the notations are well defined (e.g., defined $\log \pi(y|x) = \sum_i \log \pi(y_i|x)$), (8) should be enough.

**Questions:**

See weaknesses.

In addition, as suggested by Fig. 2, TVD would encourage the appearance of non-existing mode, i.e., it will also encourage the probability of regions with zero/low probabilities in the support of the chosen responses. Do you observe such a behavior in practice? For example, if you substitute the generated samples into a larger (ideally GPT-like) model, do you observe there will be more samples with relatively low perplexity compared to a model trained with SimPO and comparable loss?

---

> ### Author Response · Authors · 2024-11-20
> **Response to Reviewer 7YzN Part (1/2)**
>
> Dear reviewer 7YzN, we thank you for your valuable suggestions and positive feedback. We are happy to hear that you found our paper to be well-motivated and strong in both empirical and theoretical aspects. The following is our response to your comments:
>
> ---
>
> **Q1. It would be better if the experiments can show the proposed method works well across different scales of models.**
>
> **A1.** Thank you for your valuable suggestion! Following your suggestion, we conducted additional experiments on more different scales of models. Given the short rebuttal time, we choose the recent Qwen2-1.5B-Instruct  and Gemma2-9B-Instruct models. The new results are provided in Table 10 in the updated submission (the results on Qwen2-1.5B-Instruct  are provided in the table below for your reference).
>
> | Model   | AlpacaEval 2 LC |  AlpacaEval 2 WR | Arena-Hard | GSM8K  | TruthfulQA |
> |---------|--------|--------|--------|--------|------------|
> | SFT     | &nbsp; 5.42   |  &nbsp; &nbsp; 3.55   | &nbsp; 2.5    | 59.32  | 28.37      |
> | DPO     | &nbsp; 8.71   | &nbsp; &nbsp; 6.05   | &nbsp; 4.0    | 65.31  | 34.59      |
> | SimPO   | &nbsp; 10.11  | &nbsp; &nbsp;7.52   | &nbsp; 4.9    | 68.21  | 39.37      |
> | **SimPER** | &nbsp; **12.59** | &nbsp; &nbsp; **9.31** | &nbsp; **5.5** | **72.75** | **40.33** |
>
>
> These results show that SimPER continues to outperform the baselines on
> other scales of models. Combined with the comparisons involving 5 base models already presented in our paper, these results strongly validate the effectiveness and robustness of SimPER. We will include these new results in the final version.
>
> ---
>
> **Q2. In addition, I am confused about the choice of the model. It seems only Fig.4 uses Pythia-2.8B**
>
> **A2.** Thank you for your question, and we apologize for any confusion. The reason we chose Pythia-2.8B for Anthropic-HH is to strictly align all settings with those in the well-known original DPO paper and its widely-used code for Anthropic-HH, ensuring fair and reproducible comparisons. We apologize for the confusion and will clearly state this in the main text of our final version.
>
> ---
>
> **Q3. The proposed loss is fairly simple and I do believe simplicity is good. But the presentation of Eq. (5)(6)(7)(8) is redundant, and (8) should be enough.**
>
> **A3.** Thank you for your great suggestion! We included Eqs. (5), (6), and (7) to explicitly connect our simple objective with the well-known perplexity. However, we agree with the reviewer that they are somewhat redundant given SimPER's simplicity. In the final version, we will reorganize the content and remove some of these equations to enhance clarity.
>
> ---
>
> **Q4. In addition, as suggested by Fig. 2, TVD would encourage the appearance of non-existing mode, i.e., it will also encourage the probability of regions with zero/low probabilities in the support of the chosen responses. Do you observe such behavior in practice?**
>
> **A4.** Thank you for your questions! We would like to clarify the following points.
>
> As shown in Figure 2, **SimPER, which optimizes reverse TVD for mode-seeking behavior,  sharpens the probability mass on only certain high-density regions. In contrast, SimPO, which optimizes forward KLD, forces the model to learn a flat distribution that spans itself to cover all the non-zero probability regions due to its mass-covering behavior [1,2,3].**
>
> To highlight the differences between SimPO (Forward KLD) and SimPER (Reverse TVD), as well as the unique benefits of Reverse TVD, we conducted additional experiments on Anthropic-HH, chosen for its controllability. We studied how these divergences affect the trade-off between alignment and diversity in generated responses. For diversity, we generated 25 responses using top-p sampling (p=0.95) for each test prompt and measured predictive entropy and distinct-n [4].
>
> | Anthropic-HH | Alignment (Win-Rate)   | Diversity (Entropy) | Diversity (Distinct-1) | Diversity (Distinct-2) |
> | --- | --- | --- | --- | --- |
> | SimPER   | &nbsp;  59.87 |  &nbsp; 12.15  | &nbsp; 0.024    | &nbsp; 0.165 |
> | SimPO    | &nbsp;  50.33 |  &nbsp; 14.71  | &nbsp; 0.028   |  &nbsp; 0.197  |
>
> As shown in the above, SimPER’s mode-seeking property achieves significantly higher alignment performance but slight lower diversity compared to SimPO, whose mass-covering behavior results in higher diversity but lower alignment [3]. These results support our intuition: the mode-seeking nature of reverse TVD reduces diversity but is highly effective for alignment optimization, as noted in [1,3].
>
> Additionally, SimPER outperforms SimPO in challenging tasks such as math and safety alignment. One possible reason is that these tasks have few valid answers per input, making mode-seeking more beneficial. This reinforces the potential of mode-seeking for alignment, consistent with prior observations [1,3].  We will clearly state this in the main text of our submission.

---

> ### Author Response · Authors · 2024-11-20
> **Response to Reviewer 7YzN Part (2/2)**
>
> **We sincerely hope that our responses can address your comments. As noticed by the reviewer, our work presents some interesting findings, a simple yet effective framework, and some theoretical contributions. The reviewer's suggestions can be easily and effectively addressed, and we genuinely hope that the reviewer can consider increasing the score. Thank you very much for your time!**
>
> [1] Preference Fine-Tuning of LLMs Should Leverage Suboptimal, On-Policy Data. ICML 2024.
>
> [2] Learning to Generalize from Sparse and Underspecified Rewards. ICML 2019.
>
> [3] Towards Efficient Exact Optimization of Language Model Alignment. ICML 2024
>
> [4] Beyond Reverse KL: Generalizing Direct Preference Optimization with Diverse Divergence Constraints. ICLR 2024

---

> > ### Comment · Reviewer_7YzN · 2024-12-01
> > **Reviewer Response**
> >
> > Thanks for the detailed response! I have no more questions and will keep my positive score.

---

> > > ### Author Response · Authors · 2024-12-01
> > > **Thank you for your reply**
> > >
> > > Dear Reviewer 7YzN,
> > >
> > > Thank you very much for your reply and support. We sincerely appreciate your recognition of our contribution.
> > >
> > > We are truly grateful for your time and your reply.
> > >
> > > Authors

---

### Official Review · Reviewer_bH9h · 2024-11-03

**Soundness:** 3
**Presentation:** 3
**Contribution:** 4
**Rating:** 8
**Confidence:** 5

**Summary:**

This paper proposes an effective, hyperparameter-free objective for preference optimization, named SimPER. Comprehensive experiments are conducted on popular benchmarks to verify the effectiveness of SimPER.

**Strengths:**

- The proposed SimPER training objective is parameter-free, eliminating the need for hyperparameter tuning required in prior work.

- The paper provides detailed theoretical analysis and experimental results to demonstrate the superiority of SimPER.

**Weaknesses:**

I did not observe any significant flaws. However, there is still room for improvement in the details of experimental results.

**Questions:**

Response length on AlpacaEval 2 is important, so the author(s) may consider adding the average response length to the results. Additionally, as an updated version of MT-Bench, Arena-Hard provides more reliable evaluations compared to MT-Bench and is widely used in previous works of preference optimization. The author(s) might consider including Arena-Hard to better demonstrate performance.

---

> ### Author Response · Authors · 2024-11-20
> **Response to Reviewer bH9h**
>
> Dear reviewer bH9h, we sincerely appreciate your recognition of our contributions to simplicity, effectiveness, and theoretical analysis, and we are grateful for the encouraging comments. Please find our responses below:
>
> ---
> **Q1. The author(s) may consider adding the average response length to the results.**
>
> **A1.** Thank you for your great suggestion. We have added additional results for the average response length in AlpacaEval 2.0 to Table 5 in the updated submission. These new results, along with our improvement in the length-controlled (LC) win rate, demonstrate that SimPER does not significantly increase response length compared to the baselines, despite SimPER's superior performance and simplicity.
>
> ---
> **Q2. The author(s) might consider including Arena-Hard to better demonstrate performance.**
>
> **A2.** We thank the reviewer for suggesting another chat-based benchmark Arena-Hard. Following your suggestion, we have included additional comparisons on Arena-Hard in Table 5 of the updated submission (also provided in the table below). For Arena-Hard, we report the win rate (WR) against the baseline model, consistent with previous work SimPO [1]. These new results further strengthen our contribution and demonstrate that our simple yet effective SimPER achieves superior performance on Arena-Hard.
>
> | Models  | DPO  | CPO | ORPO | SimPO | SimPER |
> | --- | --- | --- | --- | --- |--- |
> | Mistral-7B-Base | 10.4 | 6.9  | 7.0 | 16.6 | **18.5** |
> | Llama3-8B- Base | 15.9 |  5.8 | 10.8  | 23.4 | **24.7** |
>
>
> ---
>
> **In light of these responses, we sincerely hope our rebuttal has addressed your comments.  If you have any other comments, please do share them with us, and we will address them further. Thank you for your time and efforts!**
>
>
> [1] SimPO: Simple Preference Optimization with a Reference-Free Reward. NeurIPS 2024.

---

### Author Response · Authors · 2024-11-20
**Global Response to Reviewers**

We sincerely thank all the reviewers for their insightful comments and helpful suggestions. We are encouraged to see that all reviewers agree our paper is "simple and effective," "well-motivated," and demonstrates "solid analysis and good empirical results."

In this work, we address the limitations of current preference alignment methods, particularly their sensitivity to hyperparameters, which poses a critical and practical  challenge in alignment. To tackle this, we propose a surprisingly effective hyperparameter-free objective for alignment through perplexity optimization, a contribution that reviewers found particularly compelling.

We have made every effort to address all comments, suggestions, and misunderstandings in our responses and have prepared a point-by-point reply for each reviewer. Below, we summarize the major responses while addressing the comments of each reviewer individually:

- We provided additional results, including comparisons with ORPO.
- We clarified several details to address some misunderstandings from the reviewers.
- We also further evaluated SimPER in the online iterative scenario suggested by Reviewer MQ2x.

We have updated the submission to include these additional results and clarifications (marked in red text).

We extend our sincere thanks to all the reviewers once again for their time and efforts.

Best regards,

Authors of “SimPER: Simple Preference Fine-Tuning without Hyperparameters by Perplexity Optimization”

---

### Meta-Review · Area_Chair_o66h · 2024-12-21

**Metareview:**

This paper introduces SimPER as a state-of-the-art method for preference optimization in LLM alignment.

- Pros: Its key contribution is a hyperparameter-free objective based on inverse perplexity, which the authors demonstrate exhibits mode-seeking behavior that effectively aligns with human preferences. Extensive experiments highlight SimPER's superior performance compared to common baselines across multiple benchmarks.

- Cons: The experiments focus primarily on 7–8B models, with limited exploration of larger scales. Additionally, while SimPER achieves strong alignment, it comes at the cost of reduced response diversity.

SimPER's simplicity, robustness, and strong empirical results make it a notable advancement in preference optimization for LLMs.

**Additional Comments On Reviewer Discussion:**

The authors’ rebuttal and additional experiments resolved all critical concerns. Reviewers unanimously maintained or raised their scores, reflecting the paper’s strong overall reception.

---

### Decision · Program_Chairs · 2025-01-22

Accept (Poster)